# Geometric Representation Condition Improves Equivariant Molecule Generation

**Zian Li** [* 1 2] **Cai Zhou** [* 3 4] **Xiyuan Wang** [1 2] **Xingang Peng** [1 2] **Muhan Zhang** [1]

## Abstract

Recent advances in molecular generative models have demonstrated great promise for accelerating scientific discovery, particularly in drug design. However, these models often struggle to generate high-quality molecules, especially in conditional scenarios where specific molecular properties must be satisfied. In this work, we introduce GeoRCG, a general framework to improve molecular generative models by integrating geometric representation conditions with provable theoretical guarantees. We decompose the generation process into two stages: first, generating an informative geometric representation; second, generating a molecule conditioned on the representation. Compared with single-stage generation, the easy-to-generate representation in the first stage guides the second stage generation toward a high-quality molecule in a goal-oriented way. Leveraging EDM and SemlaFlow as base generators, we observe significant quality improvements in unconditional molecule generation on the widely used QM9 and GEOM-DRUG datasets. More notably, in the challenging conditional molecular generation task, our framework achieves an average 50% performance improvement over state-of-the-art approaches, highlighting the superiority of conditioning on semantically rich geometric representations. Furthermore, with such representation guidance, the number of diffusion steps can be reduced to as small as 100 while largely preserving the generation quality achieved with 1,000 steps, thereby significantly reducing the generation iterations needed. Code is available at https://github.com/GraphPKU/GeoRCG.

*Equal contribution [1]Institute for Artificial Intelligence, Peking University, Beijing, China [2]School of Intelligence Science and Technology, Peking University, Beijing, China [3]Department of Electrical Engineering and Computer Science, Massachusetts Institute of Technology, Cambridge, MA, USA [4]Department of Automation, Tsinghua University, Beijing, China. Correspondence to: Muhan Zhang <muhan@pku.edu.cn>.

*Proceedings of the 42nd International Conference on Machine Learning*, Vancouver, Canada. PMLR 267, 2025. Copyright 2025 by the author(s).

## 1. Introduction

Recent years have seen rapid development in generative modeling techniques for molecule generation (Garcia Satorras et al., 2021; Hoogeboom et al., 2022; Luo & Ji, 2022; Wu et al., 2022; Xu et al., 2023; Le et al., 2023; Morehead & Cheng, 2024), which have demonstrated great promise in accelerating scientific discoveries such as drug design (Graves et al., 2020). By representing molecules as *point clouds of chemical elements* embedded in Euclidean space (potentially with edges (Vignac et al., 2023; Irwin et al., 2024)) and employing equivariant models such as EGNN (Satorras et al., 2021) as backbone architectures, these approaches ensure the *O(3)- (or SO(3)-) invariance* of the modeled molecule probability and have shown significant progress in both unconditional and conditional molecule generation tasks.

Despite the advances, precisely modeling the molecular distribution $q(\mathcal{M})$ still remains a challenge, with current models often falling short of satisfactory results. This is especially true in more practical scenarios where the goal is to capture the conditional distribution $q(\mathcal{M}|c)$ for conditional generation, with $c$ representing a desired property such as the HOMO-LUMO gap. In such cases, recent models still produce molecules with property errors *significantly larger* than the data lower bound (Hoogeboom et al., 2022; Xu et al., 2023). This challenge arises in part because molecules are naturally supported in a lower-dimensional manifold (Mislow, 2012; De Bortoli, 2022; You et al., 2023), yet they are embedded in a 3D space with much higher ambient dimensions ($N \times (3 + d)$), where $N$ is the number of atoms and $d$ the atom feature dimension). Consequently, directly learning these distributions without additional guidance or conditioning solely on a single property can result in substantial errors (Song et al., 2021), often leading to unstable or undesirable molecular samples.

In this work, we propose GeoRCG (**Geo**metric-**R**epresentation-**C**onditioned Molecule **G**eneration), a general framework for improving the generation quality of molecular generative models by leveraging *geometric representation conditions* for both unconditional and conditional generation; see Figure 1 for an overview of the framework. At a high level, rather than directly learning the extrinsic molecular distribution, we aim to

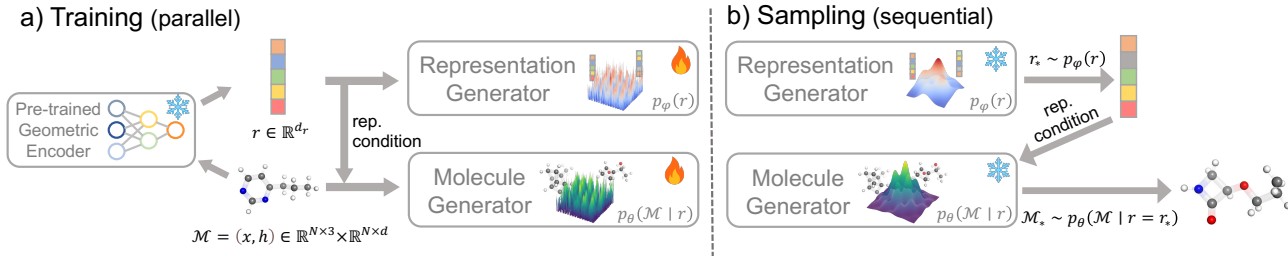

Figure 1: Training and sampling procedure of GeoRCG for unconditional molecule generation. a) During training, each molecule $\mathcal{M}$ is mapped into a representation $r$ by a pre-trained, frozen geometric encoder $E$. The representation distribution is then learned by a lightweight representation generator. The molecule generator is trained in a self-conditioned manner, generating a molecule $\mathcal{M}$ conditioned on its own representation $E(\mathcal{M})$. b) During sampling, an informative representation is first generated, which subsequently guides the molecule generator to produce high-quality molecules.

first transform it into a more compact and semantically meaningful representation distribution, with the help of a well-pretrained geometric encoder $E$ such as Unimol (Zhou et al., 2023) and Frad (Feng et al., 2023). This distribution is much simpler because it does not exhibit any group symmetries, such as $O(3)/SO(3)$ and $S(N)$ groups which are present in extrinsic molecular distributions. As a result, a lightweight representation generator (Li et al., 2023) can effectively capture this simple distribution. In the second stage, we employ a conditional molecular generator to achieve the ultimate objective: molecular generation. Unlike conventional approaches, our molecular generator is directly informed by the first-stage geometric representation, which encapsulates crucial molecular structure and property information. This guidance enables the generation of high-quality molecular structures with improved fidelity.

Our approach is directly inspired by RCG (Li et al., 2023), which, however, focuses on image data with fixed sizes and positions and does not necessitate handling Euclidean and permutation symmetries—factors that are markedly different in molecular data. Compared to recent work GraphRCG (Wang et al., 2024) which applies the RCG framework to 2D graph data, we explicitly handle 3D geometry that is more complex due to the additional Euclidean symmetry. Moreover, we avoid the complicated stepwise bootstrapped training and sampling process proposed in Wang et al. (2024) that requires noise alignment, sequential training, and simultaneous encoder training. Instead, we adopt a simple and intuitive framework that enables parallel training and leverages advanced pre-trained geometric encoders containing valuable external knowledge (Zaidi et al., 2022; Feng et al., 2023), thus achieving competitive results without complex training procedures. Notably, while Li et al. (2023) primarily focuses on empirical evaluation, we also provide *generic theoretical characterizations* of the representation-conditioned diffusion model class for both unconditional and conditional generation, offering a rigorous understanding of the improved performance.

To illustrate the effectiveness of our approach, we select one of the *simplest* and most classical equivariant generative models, EDM (Hoogeboom et al., 2022), as the base molecular generator of GeoRCG. For better performance on the more challenging dataset GEOM-DRUG, we also apply GeoRCG onto the recent state-of-the-art (SOTA) model SemlaFlow (Irwin et al., 2024). Experimentally, our method achieves the following significant improvements:

- Substantially **enhancing the quality** (e.g., molecule stability) of the generated molecules on the widely used QM9 and GEOM-DRUG datasets. On QM9, GeoRCG not only improves the performance of EDM by a large margin, but also significantly surpasses several recent baselines with advanced performance (Wu et al., 2022; Xu et al., 2023; Morehead & Cheng, 2024; Song et al., 2024a). On GEOM-DRUG, GeoRCG also significantly improves EDM's performance, and consistently enhances SemlaFlow's already SOTA results.
- More remarkably, in **conditional molecule generation tasks**, GeoRCG yields an average **50% improvement** in performance (i.e., difference of generated molecule's property with conditions), while many contemporary models struggle to achieve even marginal gains.
- By incorporating classifier-free guidance into the molecule generator (Li et al., 2023) and employing low-temperature sampling for representation generation (Ingraham et al., 2023), GeoRCG demonstrates a **flexible trade-off between molecular quality and diversity** on QM9 dataset without additional training, which is especially advantageous in specific molecular generation tasks that prioritize quality over diversity.
- With the assistance of the representation guidance, GeoRCG significantly **reduces the number of diffusion steps** required by approximately 10x, while preserving the quality of molecular generation.

## 2. Related Works

**Molecular Generative Models.** Early work has primarily focused on modeling molecules as 2D graphs (composed of atom types, connections, and edge types), utilizing 2D graph generative models to learn the graph distribution (Vignac et al., 2022; Jang et al., 2023; Le et al., 2023; Jo et al., 2023; Luo et al., 2023; Zhou et al., 2024). However, since molecules inherently exist in 3D space where physical laws govern their behavior and spatial geometry provides critical information related to key properties, recent research has increasingly focused on leveraging 3D generative models to directly learn the *geometric* distribution by modeling molecules as point clouds of chemical elements. Notable early autoregressive models include G-SchNet (Gebauer et al., 2019) and G-SphereNet (Luo & Ji, 2022). More recently, diffusion models have demonstrated effectiveness in this domain, as evidenced by models like EDM (Hoogeboom et al., 2022) and subsequent advancements that enhance EDM with latent space (Xu et al., 2023), prior information (Wu et al., 2022) and more powerful backbones (Morehead & Cheng, 2024). Furthermore, recent advances in flow methods (Lipman et al., 2022; Liu et al., 2022b) have inspired the development of geometric, equivariant flow methods including EquiFM (Song et al., 2024b) and GOAT (Hong et al., 2024), which enable much faster molecule generation speed. Beyond these, there are also methods that jointly model 2D and 3D information (Vignac et al., 2023; You et al., 2023; Huang et al., 2024; Irwin et al., 2024) (also called 3D graph (You et al., 2023)), where representative methods include MiDi (Vignac et al., 2023) and SemlaFlow (Irwin et al., 2024) that jointly learn atom types, bond types, formal charges and coordinates.

**Pre-training for Molecular Encoders** Learning meaningful molecular representations is crucial for downstream tasks like molecular property prediction (Fang et al., 2022). The strategy of pre-training on large-scale datasets followed by fine-tuning on smaller, task-specific datasets has been proven to significantly improve model performance in vision and language domains (Kenton & Toutanova, 2019; Brown, 2020; Dosovitskiy, 2020). Building on this success, recent studies have explored pre-training methods for molecular data, aiming to achieve similar performance improvements (Zhou et al., 2023; Feng et al., 2023; Liu et al., 2022a; Fang et al., 2022; Jiao et al., 2024; Ni et al., 2024). Common pretext tasks involve masking and recovering atom types, bond lengths, or bond angles (Fang et al., 2022; Zhou et al., 2023). However, since molecules exist in continuous 3D space, a more effective approach is introduced by adding carefully crafted noise into the molecular coordinates and training the model to denoise it. Examples of such noise types include isotropic Gaussian noise (Zaidi et al., 2022; Zhou et al., 2023), Riemann-Gaussian noise (Jiao et al., 2023), and complex hybrid noise (Ni et al., 2024; Feng et al., 2023; Jiao et al., 2024). Notably, Zaidi et al. (2022) showed that denoising equilibrium structures effectively corresponds to learning the underlying force field, thereby producing molecular representations that are physically and chemically informative.

**Latent Generative Models.** At a high level, our framework can also be viewed as a latent generative model, where data distributions are learned in a latent space (our stage 1) and decoded back through some decoder (our stage 2). Most prior work in this domain either focuses on regular data forms (e.g., images) with fixed positions and sizes (Van Den Oord et al., 2017; Razavi et al., 2019; Dai & Wipf, 2019; Aneja et al., 2021; Rombach et al., 2022; Li et al., 2023), or on graph data without Euclidean symmetry and requires explicit modeling (Wang et al., 2024). Molecular data, however, presents unique challenges in both aspects. One of the key issues in this context is how to define the latent space—defining it as "latent coordinates and features" as in GeoLDM (Xu et al., 2023) still results in a geometrically structured and thus complex space, while defining it on representations as we do introduces the challenge of effectively "decoding" a global, non-symmetric embedding back into geometric objects. LGD (Zhou et al., 2024) trains a diffusion model on a unified Euclidean latent space obtained by jointly training a powerful encoder and a simple decoder, and performs both generation and prediction tasks focusing on 2D graphs. LDM-3DG (You et al., 2023) adopts representation latent space but employs a cascaded (2D+3D) autoencoder (AE) framework, where the decoder is designed (or trained) to be *deterministic*, rendering poor performance on the 3D part as evidenced in our experiments. In contrast, we model the decoder as a powerful *generative model*, focusing solely on geometric learning while demonstrating superior effectiveness.

## 3. Methods

### 3.1. Preliminaries

In this work, we represent molecules as *point clouds of chemical elements* in 3D space, denoted by $\mathcal{M} = (\mathbf{x}, \mathbf{h})$, where $\mathbf{x} = (\mathbf{x}_1, \ldots, \mathbf{x}_N)^\top \in \mathbb{R}^{N \times 3}$ represents the atomic coordinates of $N$ atoms, and $\mathbf{h} = (\mathbf{h}_1, \ldots, \mathbf{h}_N)^\top \in \mathbb{R}^{N \times d}$ captures the node features of dimension $d$, such as atomic numbers and charges. This formulation follows the approach of Hoogeboom et al. (2022); Xu et al. (2023); Morehead & Cheng (2024) and is widely adopted in molecular representation learning (Thomas et al., 2018; Li et al., 2024a; Zaidi et al., 2022), facilitating the integration of pre-trained molecular encoders (Zaidi et al., 2022; Feng et al., 2023). After generating point clouds of chemical elements, these methods infer bond types using lookup tables based on atom types and pairwise distances, or relying on advanced packages

like OpenBabel (O'Boyle et al., 2011). Notably, approaches like MiDi (Vignac et al., 2023) and SemlaFlow (Irwin et al., 2024) additionally represent molecules with explicit bond types, enabling joint learning and generating of 2D and 3D information, which typically results in improved performance. We use $q$ to denote the underlying data distribution, such as molecule distributions $q(\mathcal{M})$, and $p$ to denote the approximated distributions captured by parametric models.

We denote the pre-trained geometric encoder as $E$ : $\bigcup_{N=1}^{+\infty}(\mathbb{R}^{N\times 3} \times \mathbb{R}^{N\times d}) \to \mathbb{R}^{d_r}$, which embeds a molecule $\mathcal{M}$ with an arbitrary number of nodes $N$ into a representation vector $r$ of fixed dimension $d_r$. The geometric encoder exhibits *E(3)- (or SE(3)-) invariance*, suggesting that $E(\mathcal{M}) = E(\mathbf{x}, \mathbf{h}) = E(\mathbf{x}\mathbf{R}^T + \mathbf{t}, \mathbf{h})$ for any $\mathbf{t} \in \mathbb{R}^3$ and $\mathbf{R} \in O(3)$ (or $SO(3)$), where $O(3)$ is the set of orthogonal matrices (and $SO(3)$ being the set of special orthogonal matrices).

### 3.2. GeoRCG: Geometric-Representation-Conditioned Molecular Generation

**Geometric Representation Generator.** To improve the quality of the generated molecules, we propose to first transform the geometrically structured molecular distribution $q(\mathcal{M})$ into a non-geometric representation distribution $q(r)$ using a well-pretrained geometric encoder $E$ that maps each molecule $\mathcal{M}$ to its representation $r$. Learning the representation distribution $q(r)$ is considerably easier, since representations do not exhibit any symmetry as in explicit molecular generative models (Hoogeboom et al., 2022). We thus leverage a simple yet effective MLP-based diffusion architecture as proposed in (Li et al., 2023) for the representation generator $p_\varphi(r)$, which follows DDIM schemes (Song et al., 2020a) for training and adopts predictor-corrector frameworks for sampling (Song et al., 2020b).

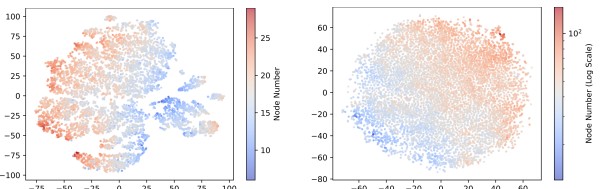

Figure 2: t-SNE visualizations of the representations produced by Frad (Feng et al., 2023) for the QM9 dataset (left) and by Unimol (Zhou et al., 2023) for the GEOM-DRUG dataset (right). The representations exhibit clear clustering based on node count.

One additional design compared to previous practices (Li et al., 2023; Wang et al., 2024) is that we condition the representation generator on the molecule's node number $N$ by default[1]. This is crucial to ensuring *consistency* between the

size of the representation's underlying molecule and the size of the molecule it guides to generate. Moreover, molecules with different sizes often have distinct modes in structures and properties (Hoogeboom et al., 2022), which is reflected in their geometric representations learned by modern pre-trained geometric encoders (Zhou et al., 2023; Feng et al., 2023), as shown in Figure 2. From the figures, it is evident that by conditioning on $N$, the learning process for the representation generator becomes simpler and more effective, leading to the following loss function of our representation generator:

$$\mathcal{L}_{\text{rep}} = \mathbb{E}_{(r,N)\in\mathcal{D}_{\text{train}}^{\text{rep}}, \epsilon\sim\mathcal{N}(0,I), t\sim\mathcal{U}(0,T)} \left[||r - f_\varphi(r_t; t, N)||^2\right],$$ (1)

where $\mathcal{D}_{\text{train}}^{\text{rep}} = \{(E(\mathcal{M}), N(\mathcal{M}))|\mathcal{M} \in \mathcal{D}_{\text{train}}^{\text{mol}}\}$, with $N(\mathcal{M})$ representing atom number of $\mathcal{M}$ and $\mathcal{D}_{\text{train}}^{\text{mol}}$ denoting the molecule dataset. Here, $f_\varphi$ is the MLP backbone (Li et al., 2023), and $r_t = \sqrt{\alpha_t}r + \sqrt{1 - \alpha_t}\epsilon$ is the noisy representation computed with the predefined schedule $\alpha_t \in (0, 1]$.

**Molecule Generator.** Since the ultimate goal of our framework is to generate molecules from $q(\mathcal{M})$, we decompose the molecular distribution as $q(\mathcal{M}) = \int q(\mathcal{M}|r)q(r)\, \mathrm{d}r$ to explicitly enable geometric-representation conditions. Consequently, a geometric-representation-conditioned molecular generator $p_\theta(\mathcal{M}|r)$ is required. In principle, we can use many modern molecule generators (Hoogeboom et al., 2022; Xu et al., 2023; Morehead & Cheng, 2024; Irwin et al., 2024), as these models can all take additional conditions.

To illustrate the effectiveness of our approach, we choose a relatively simple model EDM (Hoogeboom et al., 2022) as the base generator and *primarily demonstrate our method with it*. Furthermore, we showcase the generality of our approach by adapting it to a recent flow-matching based SOTA model, SemlaFlow (Irwin et al., 2024), emphasizing its ability to consistently improve SOTA models' performance.

EDM is designed to ensure the $O(3)$-invariance, i.e., for any $\mathbf{R} \in O(3)$, $p_\theta(\mathcal{M}) = p_\theta(\mathbf{x}, \mathbf{h}) = p_\theta(\mathbf{x}\mathbf{R}^T, \mathbf{h})$. To accommodate EDM to representation conditions, we use the following training objective:

$$\mathcal{L}_{\text{mol}} = \mathbb{E}_{(\mathcal{M},r)\sim\mathcal{D}_{\text{train}}^{\text{mol-rep}}, t\sim\mathcal{U}(0,T), \epsilon\sim\hat{\mathcal{N}}(0,\mathbf{I})} \left[||\epsilon - f_\theta(\mathcal{M}_t; t, r)||^2\right],$$ (2)

where $\mathcal{D}_{\text{train}}^{\text{mol-rep}} = \{(\mathcal{M}, E(\mathcal{M}))|\mathcal{M} \in \mathcal{D}_{\text{train}}^{\text{mol}}\}$, and sampling from $\hat{\mathcal{N}}(0, \mathbf{I})$ entails drawing $\epsilon_0 = [\epsilon_0^{(x)}, \epsilon_0^{(h)}]$ from $\mathcal{N}(0, \mathbf{I})$, adjusting $\epsilon_0^{(x)}$ by subtracting its geometric center to obtain $\epsilon^{(x)}$, and setting $\epsilon = [\epsilon^{(x)}, \epsilon_0^{(h)}]$. This ensures the zero center-of-mass property, as the distribution is defined on this subspace to ensure translation invariance (Hoogeboom et al., 2022). The noisy molecule is

---

[1] We omit the condition $N$ in our probability decompositions and mathematical derivations for statement simplicity, as its inclusion does not affect the overall framework and conclusions.

given by $\mathcal{M}_t = \alpha_t^{(\mathcal{M})}[\mathbf{x}, \mathbf{h}] + \sigma_t^{(\mathcal{M})}\epsilon$, with time-dependent schedules $\alpha_t^{(\mathcal{M})}$ and $\sigma_t^{(\mathcal{M})}$, while the diffusion backbone $f_\theta$, which is instantiated with EGNN (Satorras et al., 2021), is conditioned on $r$.

**Combining the Two Generators Together.** The representation generator $p_\varphi(r)$ and the molecule generator $p_\theta(\mathcal{M}|r)$ together model the molecular distribution $p_{\varphi,\theta}(\mathcal{M}) := \int p_\theta(\mathcal{M}|r)p_\varphi(r)\,\mathrm{d}r$, which approximates the data distribution $q(\mathcal{M}) = \int q(\mathcal{M}|r)q(r)\,\mathrm{d}r$ that we aim to capture. One notable advantage of the framework is that the decomposition enables **parallel training** of the two generators. The entire training and sampling procedure is summarized in Algorithm 1.

**Theoretical Analysis of GeoRCG.** There are several key properties of GeoRCG that facilitate high-quality molecule generation. First, GeoRCG preserves symmetry properties of the base molecule generator $p_\theta(\mathcal{M})$:

**Proposition 3.1.** *(Symmetry Preservation) Assume the original molecular generator $p_\theta(\mathcal{M})$ is O(3)- or SO(3)-invariant. Then, the two-stage generator $p_{\varphi,\theta}(\mathcal{M})$ is also O(3)- or SO(3)-invariant.*

*Proof.* This result follows directly from the definition. Specifically, $p_{\varphi,\theta}(\mathcal{M}) = \int p_\theta(\mathcal{M}|r)\,p_\varphi(r)\,\mathrm{d}r = \int p_\theta(\mathbf{x}\mathbf{R}^T, \mathbf{h}|r)\,p_\varphi(r)\,\mathrm{d}r = p_{\varphi,\theta}(\mathbf{x}\mathbf{R}^T, \mathbf{h})$ for any $\mathbf{R} \in O(3)$ (or $SO(3)$). The second equality holds due to the symmetric property of $p_\theta(\mathcal{M})$, which remains valid when additional non-symmetric conditions $r$ are applied.

Moreover, representation-conditioned diffusion models can achieve no higher overall total variation distance than traditional diffusion models, and can arguably yield better results, as the representation encodes key data information that may further reduce estimation error. We present the rigorous bound in Theorem 3.2, and provide corresponding proof and detailed discussions in Appendix D.1. Remarkably, this is a *generic theoretical characterization* that applies to prior *experimental* work (Li et al., 2023). For models that account for equivariant symmetries such as EDM, we build upon results from (Feng et al., 2024; You et al., 2023) to establish finer-grained bounds, as detailed in Theorem D.14.

**Theorem 3.2.** *Consider the random variable $x \in \mathbb{R}^{N(d+3)} \sim q(x)$, and assume that the second moment $m_x$ of $x$ is bounded as $m_x^2 := \mathbb{E}_{q(x)}[\|x - \bar{x}\|^2] < \infty$, where $\bar{x} := \mathbb{E}_{q(x)}[x]$. Further, assume that the score $\nabla \ln q(x_t)$ is $L_x$-Lipschitz for all $t$, and that the score estimation error in the second-stage diffusion is bounded by $\epsilon_{\varphi,\theta,cond}$ such that $\mathbb{E}_{r \sim p_\varphi(r),\, x_t \sim q_t(x_t|r)}[\|s_\theta(x_t, t, r) - \nabla \ln q_t(x_t|r)\|^2] \leq \epsilon_{\varphi,\theta,cond}^2$. Denote the step size as $h := T/N_d$, where $T$ is the total diffusion time and $N_d$ is the number of discretization steps, and assume that $h \preceq 1/L_x$. Suppose that we sample $x \sim p_\theta(x|r)$ from Gaussian noise, where $r \sim p_\varphi(r)$, and denote the final distribution of $x$ as $p_{\theta,\varphi}(x)$.*

*Define $p_0^{q_{T|\varphi}}$, which is the ending point of the reverse process starting from $q_{T|\varphi}$ instead of Gaussian noise. Here, $q_{T|\varphi}$ is the $T$-th step in the forward process starting from $q_{0|\varphi} := \frac{1}{A} \int_r q(x_0|r)p_\varphi(r)\,\mathrm{d}r$, where $A$ is the normalization factor. Denote the $k$-dim isotropic Gaussian distribution as $\gamma^k$. Then the following holds,*

$$\mathrm{TV}(p_{\theta,\varphi}(x), q(x)) \preceq \underbrace{\sqrt{\mathrm{KL}(q_{0|\varphi}||\gamma^{N(d+3)})}\exp(-T)}_{\textit{convergence of forward process}}$$

$$\tag{3}$$

$$+ \underbrace{(L_x\sqrt{N(d+3)h} + L_x m_x h)\sqrt{T}}_{\textit{discretization error}}$$

$$\tag{4}$$

$$+ \underbrace{\epsilon_{\varphi,\theta,\mathrm{cond}}\sqrt{T}}_{\textit{conditional score estimation error}} \tag{5}$$

$$+ \underbrace{\mathrm{TV}(q_{0|\varphi}, q_0)}_{\textit{representation generation error}} \tag{6}$$

**Balancing Quality and Diversity of Molecule Generation.** In many scientific applications, researchers prioritize generating higher-quality molecules over more diverse ones. To facilitate this, we introduce a feature that allows fine-grained control over the trade-off between diversity and quality *in the sampling stage (thus without retraining)*. This is achieved by integrating two key techniques: low-temperature sampling (Ingraham et al., 2023) (controlled via the temperature $\mathcal{T}$) for the representation generator, and classifier-free guidance (Ho & Salimans, 2022; Zheng et al., 2023) (controlled via the coefficient $w$) for the molecule generator. We provide more details about the two techniques in Appendix A. The combination of the two techniques enables flexible and explicit control, which we refer to as "Balancing Controllability" and demonstrate its effectiveness in Section 4.2.

**Handling Conditional Molecule Generation.** The framework discussed thus far focuses on unconditional molecule generation, where no specific property $c$ (e.g., HOMO energy) is prespecified. However, for molecule generation, a more practical and desired scenario is conditional (also called controllable) generation, where additional conditions $c$, such as the HOMO-LUMO gap energy, are introduced, and our objective shifts to generating molecules from the distribution $q(\mathcal{M}|c)$. In GeoRCG, this conditional generation is naturally decomposed as $p_{\theta,\varphi}(\mathcal{M}|c) := \int p_\theta(\mathcal{M}|r)p_\varphi(r|c)\,\mathrm{d}r$, suggesting that we first generate a "property-meaningful" molecular representation $r$, which is then *independently* used to condition the second-stage molecule generation; see Figure 3 for an illustration. A key advantage of this modeling approach is that, when different properties (e.g., HOMO, LUMO, GAP energy) need to be captured, **only the representation generator**

**requires retraining** under the new conditions. This retraining is highly efficient due to the lightweight nature of the representation generator. Notably, GeoRCG demonstrates outstanding conditional generation performance, as shown in Section 4.3. Moreover, we theoretically demonstrate that, under mild assumptions, the representation generator can provably estimate the conditional distribution and generate representations that lead to provable reward improvements toward the target, which subsequently benefits the second-stage generation. Further theoretical details are provided in Appendix D.2.

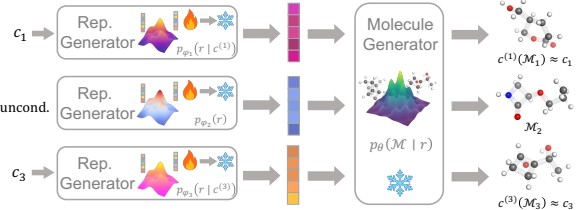

Figure 3: A single molecule generator can be employed for both unconditional and conditional molecule generation with respect to various properties. For conditional generation, only the representation generator is re-trained on (molecule, property) pairs, allowing it to conditionally sample property-meaningful representations during the sampling stage.

## 4. Experiments

### 4.1. Experiment Setup

**Datasets and Tasks.** As a method for 3D molecule generation, we evaluate GeoRCG on the widely used datasets QM9 (Ramakrishnan et al., 2014) and GEOM-DRUG (Gebauer et al., 2019; 2022; Axelrod & Gomez-Bombarelli, 2022). We focus on two tasks: unconditional molecule generation, where the goal is to sample from $q(\mathcal{M})$, and conditional (or controllable) molecule generation, where a property $c$ is given, and we aim to sample from $q(\mathcal{M}|c)$.

We use "GeoRCG (EDM)" to denote the variant of GeoRCG that employs EDM as the base molecule generator, "GeoRCG (Semla)" to refer to its application built upon SemlaFlow (Irwin et al., 2024), and "GeoRCG" when the context is clear or for general purpose. To ensure fair comparisons, we follow the dataset split and configurations exactly as in Anderson et al. (2019); Hoogeboom et al. (2022); Xu et al. (2023). Without further clarification, we **bold** the highest scores and underline the second-highest one. Additionally, to highlight the direct improvement over the base model, we display green numbers next to the score to indicate the improvement, and red numbers to denote a decrease. Without further clarification, results are calculated based on

10k randomly sampled molecules, averaged over three runs, with standard errors reported in parentheses.

**Instantiation of the Pre-trained Encoder.** We employ Frad (Feng et al., 2023), which was pre-trained on the PCQM4Mv2 dataset (Nakata & Shimazaki, 2017) using a hybrid noise denoising objective, as the geometric encoder for QM9 dataset. For GEOM-DRUG, we adopt Unimol (Zhou et al., 2023) architecture but perform our own pretraining using the dataset from (Zhou et al., 2023), with GEOM-DRUG included as an additional pretraining dataset. This is because GEOM-DRUG contains unique chemical elements not found in PCQM4Mv2 or other commonly used pretraining datasets such as ZINC or ChemBL (Li et al., 2021). We note that, when using Frad (Feng et al., 2023) as the encoder, GeoRCG also leads to significant improvements on the GEOM-DRUG dataset, although with slightly lower performance compared to Unimol; see Appendix C.

**Baselines.** A direct comparison is made with our base molecule generators, EDM or SemlaFlow. For GeoRCG (EDM), we compare it against generative models that, like EDM, do not explicitly generate bonds but instead infer them based on bond lengths. Although this approach may be less effective in generating valid molecules, it is widely adopted and *presents a greater challenge for generative models in learning 3D geometric distributions*—precisely where our geometric representation guidance offers the most significant improvement. These models include: (1) the non-equivariant counterparts of EDM and GeoLDM (Xu et al., 2023), specifically GDM(-AUG)(Hoogeboom et al., 2022) and GraphLDM(-AUG)(Xu et al., 2023); (2) the autoregressive method G-SchNet (Gebauer et al., 2019); (3) advanced equivariant diffusion models such as GeoLDM (Xu et al., 2023), EDM-Bridge (Wu et al., 2022), and GCDM (Morehead & Cheng, 2024); (4) fast equivariant flow-based methods like E-NF (Garcia Satorras et al., 2021), EquiFM (Song et al., 2024b), and GOAT (Hong et al., 2024); and (5) the recently introduced Bayesian-based method GeoBFN (Song et al., 2024a).

For GeoRCG (Semla), we compare it with recent advanced 2D&3D methods that directly generate bonds to produce higher-quality samples similar to SemlaFlow, including MiDi (Vignac et al., 2023) and EQGAT-diff (Le et al., 2023).

Note that we intentionally separate the comparison between EDM-like 3D-only models and SemlaFlow-like 2D&3D models, focusing on the improvements brought by GeoRCG to the base model. This is because combining the comparisons would be unfair, as 2D&3D models additionally learn bond information, which reduces the complexity of generating valid molecules (Morehead & Cheng, 2024).

We provide further experiments, including **ablation studies on the pre-trained encoder**, in Appendix C.

Table 1: Unconditional molecule generation on QM9 and GEOM-DRUG. The gray cells denotes the base molecule generator employed in GeoRCG.

| Methods \ Metrics | QM9 | | | | DRUG | |
|---|---|---|---|---|---|---|
| | Atom Sta (%) ↑ | Mol Sta (%) ↑ | Valid (%) ↑ | Valid & Unique (%) ↑ | Atom Sta (%) ↑ | Valid (%) ↑ |
| Data | 99 | 95.2 | 97.7 | 97.7 | 86.5 | 99.9 |
| G-Schnet | 95.7 | 68.1 | 85.5 | 80.3 | - | - |
| GDM | 97 | 63.2 | - | - | 75 | 90.8 |
| GDM-AUG | 97.6 | 71.6 | 90.4 | 89.5 | 77.7 | 91.8 |
| GraphLDM | 97.2 | 70.5 | 83.6 | 82.7 | 76.2 | 97.2 |
| GraphLDM-AUG | 97.9 | 78.7 | 90.5 | 89.5 | 79.6 | 98 |
| EDM | 98.7 | 82 | 91.9 | 90.7 | 81.3 | 92.6 |
| EDM-Bridge | 98.8 | 84.6 | 92 | 90.7 | 82.4 | 92.8 |
| GeoLDM | 98.9(0.1) | 89.4(0.5) | 93.8(0.4) | 92.7(0.5) | 84.4 | **99.3** |
| GCDM | 98.7(0.0) | 85.7(0.4) | 94.8(0.2) | 93.3(0.0) | **89** | 95.5 |
| ENF | 85 | 4.9 | 40.2 | 39.4 | - | - |
| EquiFM | 98.9(0.1) | 88.3(0.3) | 94.7(0.4) | **93.5(0.3)** | 84.1 | 98.9 |
| GOAT | 98.4 | 84.1 | 90.9 | 89.99 | 81.8 | 96.0 |
| GeoBFN | 99.08(0.03) | 90.87(0.1) | 95.31(0.1) | 92.96(0.1) | 85.6 | 92.08 |
| GeoRCG (EDM) | **99.12(0.03)** 0.43% | **92.32(0.06)** 12.59% | **96.52(0.2)** 5.03% | 92.45(0.2) 1.93% | 84.3(0.12) 3.69% | 98.5(0.12) 6.37% |

## 4.2. Unconditional Molecule Generation

We first evaluate the quality of unconditionally generated molecules from GeoRCG, with the commonly adopted validity and stability metrics for assessing molecules' quality (Hoogeboom et al., 2022). See Appendix B for detailed descriptions of these metrics.

We present the main results of GeoRCG on the QM9 and GEOM-DRUG datasets in Table 1 and Table 2. Below, we highlight the key findings: (i) **Improvement over the base model:** By leveraging geometric representations, GeoRCG significantly outperforms the base model, on both QM9 and GEOM-DRUG datasets. Notably, on QM9, GeoRCG (EDM) increases stable molecules from 82% to 93.9% and validity from 91.9% to 97.4%, while also improving molecule uniqueness. (ii) **Superior performance compared to advanced methods:** GeoRCG (EDM) also surpasses included advanced models on the QM9 dataset. On the GEOM-DRUG dataset, GeoRCG (EDM) outperforms models such as EDM-Bridge and GOAT, and gets a high score in validity. Although GeoRCG (EDM) falls short of achieving the best performance, we attribute this to the relatively limited capabilities of EDM. To address this, we replace EDM with the recent SOTA flow-matching based model, SemlaFlow (Irwin et al., 2024), as the base model on the GEOM-DRUG dataset, as shown in Table 2. As demonstrated, GeoRCG (Semla) consistently enhances SemlaFlow's SOTA performance across all metrics on the GEOM-DRUG dataset.

We proceed to investigate the "**Balancing Controllablility**" feature of GeoRCG introduced in Section 3.2. To this end, we conducted a grid search by varying both $w$ and $\mathcal{T}$ on QM9 dataset for GeoRCG (EDM), as depicted in Figure 4

Table 2: Unconditional molecule generation on GEOM-DRUG for 2D&3D methods. Molecule stability and validity are reported as percentages, while energy and strain energy are expressed in kcal·mol$^{-1}$. Results marked with $^*$ were reproduced in our own experiments.

| Methods | Atom Stab ↑ | Mol Stab ↑ | Valid ↑ | Energy ↓ | Strain ↓ |
|---|---|---|---|---|---|
| MiDi | 99.8 | 91.6 | 77.8 | - | - |
| EQGAT-diff | 99.8(0.01) | 93.4(0.21) | 94.6(0.24) | 148.8(0.9) | 140.2(0.7) |
| SemlaFlow$^*$ | **99.8(0.00)** | 97.4(0.07) | 94.4(0.17) | 95.72(1.24) | 56.42(1.07) |
| GeoRCG (Semla) | **99.8(0.00)** | **97.6(0.00)** | **95.3(0.13)** | **88.6(1.03)** | **47.64(1.10)** |

(see Appendix C for the extended figure that includes validity). The results indicate a clear trend: increasing $w$ and decreasing $\mathcal{T}$ improve validity and stability at the expense of uniqueness, allowing for fine-grained, flexible control over molecule generation. At its best, this approach achieves a molecule stability of 93.9% and a validity of 97.42%, approaching the dataset's upper bound, with a trade-off in lower validity&uniqueness of 86.82%.

Figure 4: Balance controllable generation on QM9 of GeoRCG (EDM). Increasing $w$ and decreasing $\mathcal{T}$ enhances stability, with the cost of a reduction in uniqueness.

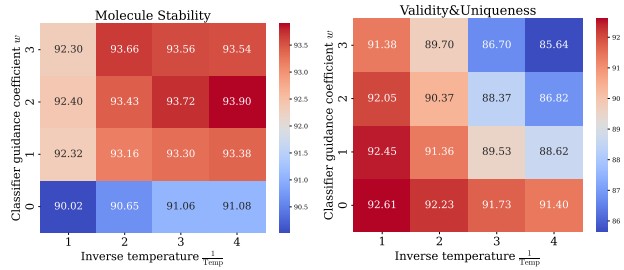

In Appendix C, we present additional experiments on QM9,

Table 3: Conditional molecule generation on QM9. The metric used is the MSE between the target property value and the classifier-predicted value. The gray cells denote the baseline molecule generator employed in our proposed approach. Models marked with ∗ indicate results obtained from our own experiments; these are provided only as a coarse reference due to potentially differing evaluation criteria, see Appendix B for details.

| Properties Methods | $\alpha$ | $\Delta\varepsilon$ | $\varepsilon_{HOMO}$ | $\varepsilon_{LUMO}$ | $\mu$ | $C_v$ |
|---|---|---|---|---|---|---|
| QM9 (lower bound) | 0.1 | 64 | 39 | 36 | 0.043 | 0.04 |
| Random | 9.01 | 1470 | 646 | 1457 | 1.616 | 6.857 |
| N_atoms | 3.86 | 866 | 426 | 813 | 1.053 | 1.971 |
| EDM | 2.76 | 655 | 356 | 584 | 1.111 | 1.101 |
| GeoLDM | 2.37 | 587 | 340 | 522 | 1.108 | 1.025 |
| GCDM | 1.97 | 602 | 344 | 479 | 0.844 | 0.689 |
| EquiFM | 2.41 | 591 | 337 | 530 | 1.106 | 1.033 |
| GOAT | 2.74 | 605 | 350 | 534 | 1.01 | 0.883 |
| LDM-3DG∗ | 12.29 | 1160 | 583 | 1093 | 1.42 | 5.74 |
| GeoBFN | 2.34 | 577 | 328 | 516 | 0.998 | 0.949 |
| GeoRCG (EDM) | **0.86(0.01)** 68.84% | **325.2(3.4)** 50.35% | **202.2(1.2)** 43.20% | **257.9(5.5)** 55.84% | **0.805(0.006)** 27.54% | **0.475(0.005)** 56.86% |

demonstrating that GeoRCG **enhances distribution-level geometric metrics**, such as BondAngleW1, which underscore GeoRCG's improved geometric learning capabilities.

### 4.3. Conditional Molecule Generation

We now turn to a more challenging task: generating molecules with a specific property value $c$ from $q(\mathcal{M}|c)$. We strictly follow the evaluation protocol outlined in (Hoogeboom et al., 2022). Specifically, QM9 is split into two halves, and an EGNN classifier (Satorras et al., 2021) is trained on the first half for evaluating the generated molecules' property, while the generator is trained on the second half. We focus on six properties: polarizability ($\alpha$), orbital energies ($\varepsilon_{HOMO}$, $\varepsilon_{LUMO}$), their gap ($\Delta\varepsilon$), dipole moment ($\mu$), and heat capacity ($C_v$).

The results are presented in Table 3. The first three baselines, as introduced by EDM (Hoogeboom et al., 2022), represent the classifier's inherent bias as the lower bound for performance, the random evaluation result as the upper bound, and the dependency of properties on $N$. For more details, please refer to Appendix B.

As shown, GeoRCG (EDM) *nearly doubles the performance* of the best existing models for most properties, with an average 50% improvement over the best ones. This is a task where many recent models struggle to make even modest improvements, as evidenced in the table. Notably, for different properties, we *only re-train the representation generator*, as demonstrated in Section 3.2, significantly saving training time. In Figure 5, we visualize the generated samples, which exhibit minimal property errors and display a clear trend as the target values increase. Additional randomly generated molecules are provided in Appendix E.2.

A potential concern is that for a given property value $c$, $p_\varphi(r|c)$ may produce a representation corresponding to a

molecule from the training dataset, allowing the molecule generator to simply recover its full conformation based on that representation. This could lead to small property errors but a lack of novelty. To address this, we conducted a thorough evaluation of the generated molecules across each property, finding that the novelty (the proportion of new molecules not present in the training dataset) remains comparable to other methods. Additionally, the conditionally generated molecules demonstrate much higher molecule stability than EDM (Hoogeboom et al., 2022). Further details can be found in Appendix C.

Table 4: Unconditional molecule generation on QM9 with fewer diffusion steps. The blue cells indicate the highest value among methods with the same number of diffusion steps, while **bold** font emphasizes values that outperform *all* other methods across all diffusion steps.

| Metrics Methods | # Steps | Atom Sta (%) ↑ | Mol Sta (%) ↑ | Valid (%) ↑ |
|---|---|---|---|---|
| Data | - | 99 | 95.2 | 97.7 |
| EquiFM | 200 | 98.9(0.1) | 88.3(0.3) | 94.7(0.4) |
| GOAT | 90 | 98.4 | 84.1 | 90.9 |
| EDM | 50 | 97.0(0.1) | 66.4(0.2) | - |
| EDM-Bridge | 50 | 97.3(0.1) | 69.2(0.2) | - |
| GeoBFN | 50 | 98.28(0.1) | 85.11(0.5) | 92.27(0.4) |
| GeoRCG (EDM) | 50 | 98.75(0.05) 1.80% | 89.08(0.52) 34.16% | 95.05(0.33) |
| EDM | 100 | 97.3(0.1) | 69.8(0.2) | - |
| EDM-Bridge | 100 | 97.9(0.1) | 72.3(0.2) | - |
| GeoBFN | 100 | 98.64(0.1) | 87.21(0.3) | 93.03(0.3) |
| GeoRCG (EDM) | 100 | **99.08(0.03)** 1.83% | **91.85(0.34)** 31.59% | **96.49(0.27)** |
| EDM | 500 | 98.5(0.1) | 81.2(0.1) | - |
| EDM-Bridge | 500 | 98.7(0.1) | 83.7(0.1) | - |
| GeoBFN | 500 | 98.78(0.8) | 88.42(0.2) | 93.35(0.2) |
| GeoRCG (EDM) | 500 | **99.09(0.01)** 0.60% | **91.89(0.24)** 13.17% | **96.57(0.12)** |
| EDM | 1000 | 98.7 | 82 | 91.9 |
| EDM-Bridge | 1000 | 98.8 | 84.6 | 92 |
| GeoBFN | 1000 | 99.08(0.06) | 90.87(0.2) | 95.31(0.1) |
| GeoRCG (EDM) | 1000 | **99.12(0.03)** 0.43% | **92.32(0.06)** 12.59% | **96.52(0.2)** 5.03% |

### 4.4. Fewer-Step Generation

With geometric representation condition, it is reasonable to expect that fewer discretization steps of the reverse diffu-

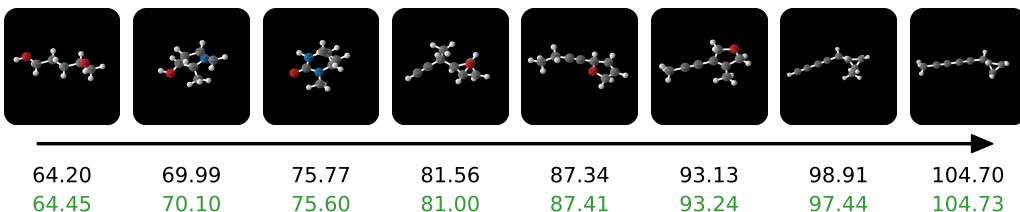

| 64.20 | 69.99 | 75.77 | 81.56 | 87.34 | 93.13 | 98.91 | 104.70 |
| 64.45 | 70.10 | 75.60 | 81.00 | 87.41 | 93.24 | 97.44 | 104.73 |

Figure 5: Conditionally generated molecules on property $\alpha$ using GeoRCG (EDM). The black number indicates the condition value, the green number represents the oracle property value for the generated molecule conformer.

Table 5: Unconditional molecule generation on GEOM-DRUG with fewer diffusion steps.

| # Steps | Atom Sta (%) ↑ | | Valid (%) ↑ | |
| --- | --- | --- | --- | --- |
| | GeoBFN | GeoRCG (EDM) | GeoBFN | GeoRCG (EDM) |
| 50 | 75.11 | **81.44(0.10)** | 91.66 | **95.70(0.70)** |
| 100 | 78.89 | **83.02(0.06)** | 93.05 | **96.30(0.70)** |
| 500 | 81.39 | **84.03(0.37)** | 93.47 | **97.57(0.90)** |
| 1000 | 85.6 | 84.3(0.12) | 92.08 | **98.5(0.12)** |

Table 6: Sampling time (in *seconds*) for 5k samples using SemlaFlow and GeoRCG (Semla) across different numbers of sampling steps, measured on a single NVIDIA RTX 4090.

| # Steps | Method | Rep. Time ↓ | Mol. Time ↓ | Mol. Time w/o CFG ↓ |
| --- | --- | --- | --- | --- |
| 100 | SemlaFlow | - | **610** | 610 |
| | GeoRCG (Semla) | 97 | 1481 | 770 |
| 50 | SemlaFlow | - | **310** | 310 |
| | GeoRCG (Semla) | 97 | 690 | 380 |
| 20 | SemlaFlow | - | **152** | 152 |
| | GeoRCG (Semla) | 97 | 315 | 189 |

sion SDE (Song et al., 2021) would still yield competitive results. Therefore, we reduce the number of diffusion steps and evaluate the model's performance. The results are presented in Table 4 and Table 5. We provide the fewer-step performance of GeoRCG (Semla) on GEOM-DRUG dataset in Appendix C.

As demonstrated, with the geometric representation condition, GeoRCG consistently outperforms other approaches across almost all step numbers. Notably, in Table 4, with approximately *100* steps, the performance of our method *nearly converges* to the optimal performance observed with 1000 steps, which already *surpasses all other methods* across all step numbers. This demonstrates the strong potential of GeoRCG to reduce the number of iterations required by sequential generative methods.

## 5. Conclusions and Limitations

**Conclusions.** In this work, we present GeoRCG, a simple yet effective framework to improve the generation quality of arbitrary molecule generators by incorporating geometric representation conditions. We use EDM (Hoogeboom et al., 2022) and SemlaFlow (Irwin et al., 2024) as base generators and demonstrate the effectiveness of our framework through extensive molecular generation experiments. In conditional generation tasks, GeoRCG achieves a remarkable 50% performance boost compared to recent SOTA models. Additionally, the representation guidance enables sampling with 10x fewer diffusion steps while maintaining near-optimal performance. Beyond these empirical improvements, we provide theoretical characterizations of representation-conditioned generative models, which address a key gap in the existing empirical literature (Li et al., 2023).

**Limitations.** We discuss two limitations of GeoRCG. First, as a representation-guided generative method, its generation quality may depend heavily on the quality of representations. For instance, in the GEOM-DRUG dataset, an insufficiently pre-trained encoder may produce less meaningful representations. As a result, the benefits of low-temperature sampling and classifier-free guidance in enhancing generation quality and controllability may be less pronounced. Future work could investigate more effective pre-training strategies beyond standard denoising or enhanced representation regularization techniques to mitigate this issue. Second, although the additional conditioning module introduces small overhead, the use of classifier-free guidance requires doubling the batch size, resulting in roughly twice the resource consumption (memory and computation), see Table 6. Nonetheless, for many cases such as GeoRCG (EDM) in QM9, performance gains are substantial even without employing classifier-free guidance. With ongoing advancements in hardwares and infrastructures, we expect the overhead introduced by parallelism to be further minimized.

## Acknowledgement

This work is supported by the National Key R&D Program of China (2022ZD0160300) and National Natural Science Foundation of China (62276003).

## Impact Statement

This paper presents work whose goal is to advance the field of molecule generation. There are many potential societal consequences of molecule generation improvement, such as accelerating drug discovery, and developing new material.

None of them we feel need to be specifically highlighted here for potential risk.

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

# A. Algorithms

**High-level Algorithm for Parallel Training and Sequential Sampling**    We provide the high-level training and sampling algorithm for GeoRCG in Algorithm 1.

---

**Algorithm 1** Parallel Training and Sequential Sampling for GeoRCG

---

**Input:** Molecule dataset $\mathcal{D}_{\text{train}}^{\text{mol}} \subset \bigcup_{N=1}^{+\infty} \left( \mathbb{R}^{N \times 3} \times \mathbb{R}^{N \times d} \right)$, pre-trained geometric encoder $E$, initial representation generator $p_{\varphi_0}(r)$, molecule generator $p_{\theta_0}(\mathcal{M}|r)$.
**Output:** Trained representation generator $p_{\varphi}(r)$, molecule generator $p_{\theta}(\mathcal{M}|r)$, and molecule samples from $p_{\varphi,\theta}(\mathcal{M})$.
**Parallel Training:**
Pre-process to obtain:
  - The representation dataset $\mathcal{D}_{\text{train}}^{\text{rep}} = \{(E(\mathcal{M}), N(\mathcal{M})) | \mathcal{M} \in \mathcal{D}_{\text{train}}^{\text{mol}}\}$
  - The mol-rep dataset $\mathcal{D}_{\text{train}}^{\text{mol-rep}} = \{(E(\mathcal{M}), \mathcal{M}) | \mathcal{M} \in \mathcal{D}_{\text{train}}^{\text{mol}}\}$
Train the representation generator $p_{\varphi_0}(r)$ with $\mathcal{D}_{\text{train}}^{\text{rep}}$ using loss $\mathcal{L}_{\text{rep}}$ in Equation (1).
Train the molecule generator $p_{\theta_0}(\mathcal{M}|r)$ with $\mathcal{D}_{\text{train}}^{\text{mol-rep}}$ using loss $\mathcal{L}_{\text{mol}}$ in Equation (2), while applying the training techniques outlined below, including representation perturbation and representation loss.
**Sequential Sampling:**
Sample a representation $r_* \sim p_{\varphi}(r)$ with low-temperature sampling technique outlined below.
Sample a molecule $\mathcal{M}_* \sim p_{\theta}(\mathcal{M}|r_*)$ conditionally, with classifier-free guidance technique outlined below.
**Return:** Trained representation generator $p_{\varphi}(r)$, molecule generator $p_{\theta}(\mathcal{M}|r)$, and generated molecule sample.

---

**Training: Representation Perturbation**    Unlike typical conditional training scenarios, GeoRCG faces a unique challenge: the representations that condition the molecule generator during training may not always coincide with those generated by the representation generator during the sampling stage. This issue is *particularly pronounced* in molecular generation than image case (Li et al., 2023), where pre-trained encoders are typically not trained on that large datasets with advanced regularization techniques like MoCo v3 (Chen et al., 2021). Consequently, the molecule generator is susceptible to *overfitting* to the training representations, as evidenced by our preliminary experiments on QM9 molecule generation shown in Table 7.

Table 7: Quality of molecules generated by GeoRCG trained on the QM9 dataset without using the representation perturbation technique, comparing different representation sources. "Training Dataset" refers to representations sampled from $\mathcal{D}_{\text{train}}^{\text{rep}}$, while "Rep. Sampler" refers to representations generated by the trained representation generator $p_{\varphi}(r)$.

| Rep source \ Metrics | Mol Sta (%) ($\uparrow$) | Valid (%) ($\uparrow$) |
|---|---|---|
| Training Dataset | 93.20 (0.50) | 97.07 (0.32) |
| Rep. Sampler | 86.93 (0.50) | 89.12 (0.21) |

We find that a simple technique—perturbing the geometric representation during training the molecule generator with some Gaussian noise $\sigma_{\text{rep}}\epsilon$, where $\epsilon \sim \mathcal{N}(0, I)$ and $\sigma_{\text{rep}}$ is a relatively small variance—is particularly effective for solving this problem. Formally, after sampling a data point $(E(\mathcal{M}), \mathcal{M})$ from $\mathcal{D}_{\text{train}}^{\text{mol-rep}}$, we use $(\mathcal{M}, E(\mathcal{M}) + \sigma_{\text{rep}}\epsilon)$ for training. Ablation study in Appendix C show this simple method can effectively prevent overfitting and ensure that performance on novel representations matches those from the training dataset.

In practice, we set $\sigma_{\text{rep}}$ to 0.3 for QM9 dataset and $\sigma_{\text{rep}}$ to [0.3, 0.5] for GEOM-DRUG dataset.

**Training: Representation Loss**    Training molecular generative models typically involves predicting a clean molecule from a noisy molecule input (in noise parameterization or vector field parameterization, an equivalent formulation exists for constructing clean molecule predictions). The loss is computed by minimizing the distance (e.g., MSE for coordinates) between the predicted clean molecule and the true clean molecule. To further strengthen supervision of the representation, we introduce an additional representation loss during training. This loss is defined as the MSE between the predicted clean molecule's representation and the actual clean molecule's representation.

In practice, for GeoRCG (EDM), we do not apply this technique, whereas for GeoRCG (Semla), we incorporate it with a relatively small coefficient.

**Sampling: Low-Temperature Sampling**    We adopt the low-temperature sampling algorithm introduced by Chroma (Ingraham et al., 2023) to the representation generator. However, we apply it to an MLP-based diffusion model rather than the equivariant diffusion model that processes geometric objects as Chroma.

The objective of low-temperature sampling is to perturb the learned representation distribution $p_\varphi(r)$ by rescaling it with an inverse temperature factor, $\frac{1}{\mathcal{T}}$, where $\mathcal{T}$ is a tunable temperature parameter during sampling, and finally enables sampling from $\mathcal{Z}_\mathcal{T} p_\phi^{\frac{1}{\mathcal{T}}}$, where $\mathcal{Z}_\mathcal{T}$ is a normalization constant. The method proposed in Chroma (Ingraham et al., 2023) scales the score $\epsilon_t$ estimated at each diffusion time step using a time-dependent factor $\lambda_t$. The approach is derived from and has theoretical guarantees for simplified toy distributions, and its performance on complex distributions, though lacking strict guarantees, has shown consistent results when combined with annealed Langevin sampling (Song et al., 2021). Here we briefly introduce it for self-containess, and recommend the readers to Ingraham et al. (2023) for detailed derivation and illustration.

Consider the vanilla reverse SDE used in DDPM sampling (VP formulation) (Song et al., 2021):

$$dr = -\frac{1}{2}\beta_t r - \beta_t \nabla_r \log q_t(r) dt + \sqrt{\beta_t} d\bar{\mathbf{w}}, \tag{7}$$

where $\bar{\mathbf{w}}$ is a reverse-time Wiener process, $q_t(r)$ denotes the ground-truth representation distribution at time $t$, and $\beta_t$ represents the time-dependent diffusion schedule. To incorporate low-temperature sampling, we utilize the following Hybrid Langevin Reverse-time SDE:

$$dr = -\frac{1}{2}\beta_t r - \left(\lambda_t + \frac{\lambda_0 \psi}{2}\right) \beta_t \nabla_r \log q_t(r) dt + \sqrt{\beta_t(1 + \psi)} d\bar{\mathbf{w}}, \tag{8}$$

where $\lambda_t$ is a time-dependent temperature parameter defined as $\frac{\lambda_0}{\alpha_t^2 + (1-\alpha_t^2)\lambda_0}$, with $\lambda_0 = \frac{1}{\mathcal{T}}$. $\alpha_t$ satisfies $\frac{1}{2}\beta_t = \frac{d \log \alpha_t}{dt}$. The parameter $\psi$ controls the rate of Langevin equilibration per unit time, and as shown in Ingraham et al. (2023), it helps align more effectively with the reweighting objective in complex distributions. In our implementation, we employ the explicit annealed Langevin process (the corrector step from (Song et al., 2021)) to achieve similar results.

In practice, for the unconditional QM9 and GEOM-DRUG generation results of GeoRCG (EDM) shown in Table 1, we set $\mathcal{T} = 1.0$ for QM9 and $\mathcal{T} = 0.5$ for GEOM-DRUG. For conditional generation in Table 3, we set $T = 1.0$. The effect of varying $\mathcal{T}$ on QM9 is detailed in Table 10. For GeoRCG (SemlaFlow) results in Table 2, we use $\mathcal{T} = 1.0$.

**Sampling: Classifier-Free Guidance**    We employ the classifier-free guidance algorithm, as introduced in (Ho & Salimans, 2022), for our molecule generator. Specifically, we introduce a trainable "fake" representation, denoted as $l$, which serves as the unconditional signal. During the training phase, $l$ is initialized as learnable parameters, and with a probability of $p_{\text{fake}}$, the true representation $r$ is replaced by $l$. This ensures that the model is capable of generating molecules unconditionally, i.e., $p_\theta(\mathcal{M}|l)$ approximates $q(\mathcal{M})$. During sampling, the final score estimate produced by the molecule generator is adjusted using the formula $(1 + w)f_\theta(\mathcal{M}_t, t, r) - wf_\theta(\mathcal{M}_t, t, l)$, allowing flexible control over the strength of the representation guidance.

In practice, for unconditional generation in Table 1, we set $w = 1.0$ for QM9 and $w = 0.0$ for GEOM-DRUG. The impact of varying $w$ on QM9 is shown in Table 10. For conditional generation in Table 3, we set $w = 2.0$. For GeoRCG (SemlaFlow), we use $w = -0.9$ (note that $-1.0 < w < 0.0$ indicates subtle representation guidance that exerts a small but still meaningful influence on the model's behavior). Further tuning of these hyperparameters may yield improved performance.

## B. Experiment Details

**Metrics and Baseline Descriptions**    We adopt the evaluation metrics, guidelines and baselines commonly used in prior 3D molecular generative models to ensure a fair comparison (Hoogeboom et al., 2022).

- In the unconditional setting, we assess the generated molecules using several key metrics:
    - **Atom Stability**: The proportion of atoms with correct valency.
    - **Molecule Stability**: The proportion of molecules where all atoms within the molecule are stable.
    - **Validity**: The proportion of molecules that can be converted into valid SMILES using RDKit.

- **Validity & Uniqueness**: The proportion of unique molecules among the valid molecules.
- **Energy**: The energy $U(x)$ of a conformation $x$. Lower energy values typically correspond to more stable and physically plausible conformations that are closer to what would be observed in nature. Following (Irwin et al., 2024), the energy is calculated using the MMFF94 force field within RDKit, a commonly used molecular modeling framework.
- **Strain**: A measure of how distorted a generated conformation $x$ is compared to its relaxed (optimized) state $\tilde{x}$. The relaxed conformation $\tilde{x}$ is obtained by applying energy minimization using the MMFF94 force field, where the molecular structure is iteratively adjusted to reduce its energy until reaching a local minimum. Mathematically, the strain energy is defined as $U(x) - U(\tilde{x})$, where $U(x)$ is the energy of the generated conformation and $U(\tilde{x})$ is the energy of the relaxed conformation. Lower strain energy values imply that the generated conformations are closer to being physically accurate and require minimal correction during optimization.

Following the approach of Hoogeboom et al. (2022); Vignac & Frossard (2021), we do not report **Novelty** scores in the main text, since QM9 represents an exhaustive enumeration of molecules satisfying a predefined set of constraints, therefore, "novel" molecule would often violate at least one of these constraints, which indicates that a model fails to fully capture the properties of the dataset. For reference, we observe that the novelty of GeoRCG (EDM) on QM9 is approximately 42% when $w = 0.0, T = 1.0$, compared to about 65% for EDM.

When comparing with 2D & 3D models in Table 8, we evaluate two 3D metrics introduced by MiDi (Vignac et al., 2023), which directly assess the geometry learning ability:

- **BondLengthW1**: The weighted 1-Wasserstein distance between the bond-length distributions of the generated molecules and the training dataset, with weights corresponding to different bond types. Formally, it is defined as:

$$\text{BondLengthsW1} = \sum_{y \in \text{bond types}} q^Y(y) \mathcal{W}1(\hat{D}_{\text{dist}}(y), D_{\text{dist}}(y)), \tag{9}$$

where $q^Y(y)$ is the proportion of bonds of type $y$ in the training set, $\hat{D}_{\text{dist}}(y)$ is the generated distribution of bond lengths for bond type $y$, and $D_{\text{dist}}(y)$ is the corresponding distribution from the test set.
- **BondAngleW1**: The weighted 1-Wasserstein distance between the atom-centered angle distributions of the generated molecules and the training dataset, with weights based on atom types. Formally, it is defined as

$$\text{BondAnglesW1} = \sum_{x \in \text{atom types}} q^X(x) \mathcal{W}1(\hat{D}_{\text{angles}}(x), D_{\text{angles}}(x)), \tag{10}$$

where $q^X(x)$ denotes the proportion of atoms of type $x$ in the training set, restricted to atoms with two or more neighbors, and $D_{\text{angles}}(x)$ represents the distribution of geometric angles of the form $\angle(r_k - r_i, r_j - r_i)$, where $i$ is an atom of type $x$, and $k$ and $j$ are neighbors of $i$.

- In the conditional generation setting, as described in (Hoogeboom et al., 2022), we evaluate our approach on the QM9 dataset across six properties: polarizability $\alpha$, orbital energies $\varepsilon_{\text{HOMO}}$, $\varepsilon_{\text{LUMO}}$, and their gap $\Delta\varepsilon$, dipole moment $\mu$, and heat capacity $C_v$. The generative model is trained conditionally on the second half of the QM9 dataset, and an EGNN (Satorras et al., 2021) classifier, trained on the first half, is employed to evaluate the *MAE* property error of the generated samples.

Three baselines are adopted in Table 3:

- **QM9 (lower bound)**: The mean error of a classifier trained on the first half of the QM9 dataset and evaluated on the second half. This baseline represents the inherent bias/error of the classifier, setting a lower bound for model performance and reflecting the best possible performance a model can achieve.
- **Random**: The classifier's performance when evaluated on the second half of QM9 with randomly shuffled molecule property labels. This baseline provides an upper bound, representing the worst achievable performance.
- **N_atoms**: The performance of a classifier trained exclusively on the number of atoms $N$ and evaluated using only $N$ as input. This baseline captures the intrinsic relationship between molecular properties and the number of atoms, which a generative model must surpass to demonstrate effectiveness.

**Model Architectures, Hyperparameters and Training Details**

- **Representation Generator.** We use the same architecture for the representation generator as the MLP-based diffusion model proposed in Li et al. (2023). We use 18 blocks of residual MLP layers with 1536 hidden dimensions, 1000 diffusion steps, and a linear noise schedule for $\beta_t$. The representation generator is trained for 2000 epochs with a batch size of 128 for both the QM9 and GEOM-DRUG datasets. Training on QM9 takes approximately 2.5 days on a single Nvidia 4090, while training on GEOM-DRUG takes around 4 days on a single Nvidia A800. Training time can indeed be further reduced, as the model shows minimal progress after approximately half of the reported time.

- **Molecule Generator.** We adopt EDM (Hoogeboom et al., 2022) as the base molecule generator, using the same EGNN (Satorras et al., 2021) architecture, with the exception of the conditioning module. Specifically, we introduce a simple gated feedforward layer to incorporate the representation condition, inserting it between each EGNN block to enhance regularization and improve model expressiveness.

  For the EGNN hyperparameters, we use 9 layers with 256 hidden dimensions for QM9 and 4 layers with 256 hidden dimensions for GEOM-DRUG. The number of diffusion steps is set to 1000 (except for cases in Table 4 that generate molecules with fewer steps), and we employ the polynomial scheduler for $\alpha_t^{(\mathcal{M})}$. Notably, all model hyperparameters are identical to those in EDM for fair comparison.

  During training, we use a batch size of 128 and 3000 epochs on QM9, and a batch size of 64 and 20 epochs on GEOM-DRUG. Training takes approximately 6 days on QM9 using a single Nvidia 4090, and around 10 days on GEOM-DRUG using two Nvidia A800 GPUs.

  For GeoRCG (SemlaFlow), we select the better-performing architecture between a gated feedforward layer and an AdaLN-Zero-like module (Peebles & Xie, 2023) for conditioning on representations and place it between every block of Semla. During training, we use the batch cost 2048 and trains for 300 epochs.

**Evaluation of LDM-3DG (You et al., 2023)**  We evaluate the performance of LDM-3DG (You et al., 2023) in Table 8 and Table 3, an Auto-Encoder-based method that also leverages the compactness of the representation space to achieve good performance.

- For the unconditional results in Table 8, we utilize the 3D conformations unconditionally generated by LDM-3DG (You et al., 2023) and compute the bond information using the look-up table method from EDM (Hoogeboom et al., 2022). Notably, although LDM-3DG predicts both the 2D molecular graph and the 3D conformation, *we do not use the bond information it predicts* for the following reasons:

  1. For the calculation of 3D geometry statistics, we observe significant inconsistencies between the generated 2D graphs and 3D geometries (e.g., valid molecules with bond lengths exceeding 100 m), leading to unreliable statistics (e.g., BondLengthW1 exploding to 3900).
  2. For stability and validity metrics, which are fundamentally 2D and computed based on molecular graphs (atoms and bond types), using the generated 2D graph would ignore the contribution of the 3D module, preventing an evaluation of its 3D learning performance.
  3. Most critically, their 2D module is explicitly designed to *filter out* invalid (sub-)molecules during generation using the RDKit method. This means that if invalid molecules or sub-molecules are generated, they are regenerated. This explicit filtering deviates from our standard evaluation criteria and is unsuitable for a fair comparison.

- For the conditional results in Table 3, we first note a potential issue with LDM-3DG (You et al., 2023): The model cannot explicitly specify the node number $N$ during molecule generation, as it uses an auto-regressive 2D generator that automatically stops adding atoms/motifs when deemed sufficient. However, the evaluation in Table 3 requires specifying both $N$ and property $c$, following the ground-truth distribution $q(N, c)$ from the training dataset. To ensure fair evaluation, conditions feeding to LDM-3DG must also satisfy this distribution. As the authors claim the model can implicitly learn $q(N)$ and thus $q(N|c)$, we first sample 10,000 values from $q(c)$ and feed them to LDM-3DG, expecting it to infer $N$ from $c$ implicitly as argued, and thus matching the $q(N, c)$ conditions.

## C. Additional Experiment Results

**Comparison of GeoRCG (EDM) with 2D&3D Methods**  We compare GeoRCG (EDM) with recent 2D&3D methods such as MiDi (Vignac et al., 2023) and LDM-3DG (You et al., 2023). As discussed in Section 4.1, such comparison is not

fair, since these models learn and generate both 2D bond structures and 3D geometries, which is beneficial for metrics like validity and stability. Considering that they rely on external chemistry toolkits like RDKit or OpenBabel (O'Boyle et al., 2011) for bond determination at the input stage and continue to leverage this bond information throughout training and generation, we report GeoRCG (EDM) using the same external tools for accurate bond computation in the generated 3D conformations, rather than relying on simple look-up tables, to narrow the comparison gap (though not completely addressed since GeoRCG (EDM) still do not explicitly learn to generate bonds).

Furthermore, as GeoRCG (EDM) essentially captures 3D geometric distributions, we place more emphasis on 3D metrics that *directly evaluate 3D learning capabilities*, including *BondLengthW1* and *BondAngleW1* proposed by Vignac et al. (2023) and detailed in Appendix B.

The results in Table 8 demonstrate that GeoRCG not only significantly outperforms MiDi and LDM-3DG on 3D metrics, highlighting the advantages of using a pure 3D model for learning 3D structures, but also further enhances EDM's performance, which has already shown considerable promise in 3D learning.

Table 8: 3D geometry statistics and generated molecule quality on QM9 across different methods. Models marked with * indicate results obtained from our own experiments; see Appendix B for the evaluation guidelines. The stability metrics for EDM are higher than in Table 1 due to using the MiDi codebase for evaluation, which permits more valency for atoms.

| Metrics / Methods | Angles (°) ↓ | Bond Length (e-2 Å) ↓ | Mol Sta (%) ↑ | Atom Sta (%) ↑ | Validity (%) ↑ | Uniqueness (%) ↑ |
|---|---|---|---|---|---|---|
| Data | ∼0.1 | ∼0 | 98.7 | 99.8 | 98.9 | 99.9 |
| MiDi (uniform) | 0.67(0.02) | 1.6(0.7) | 96.1(0.2) | 99.7(0.0) | 96.6(0.2) | 97.6(0.1) |
| MiDi (adaptive) | 0.62(0.02) | 0.3(0.1) | 97.5(0.1) | 99.8(0.0) | 97.9(0.1) | 97.6(0.1) |
| LDM-3DG* | 3.56 | 0.2 | 94.03 | 99.38 | 94.89 | 97.03 |
| EDM | 0.44 | 0.1 | 90.7 | 99.2 | 91.7 | **98.5** |
| EDM + OBabel | 0.44 | 0.1 | 97.9 | 99.8 | 99.0 | **98.5** |
| GeoRCG (EDM) | 0.21(0.04) 52.27% | **0.04(0.0)** 60% | 95.82(0.16) 5.6% | 99.59(0.02) 0.39% | 96.54(0.27) 5.28% | 95.74(0.18) 2.8% |
| GeoRCG (EDM) + OBabel | **0.20(0.04)** 54.55% | 0.07(0.06) 30% | **98.21(0.09)** 0.32% | **99.88(0.00)** 0.08% | 99.0(0.04) 0.0% | 95.74(0.16) 2.8% |

**Balancing Controllability**   We present a more comprehensive figure that includes molecule stability, atom stability, validity, and validity&uniqueness in Figure 6.

Figure 6: Balance controllable (unconditional) generation on QM9 dataset of GeoRCG (EDM). Increasing $w$ and decreasing $\mathcal{T}$ enhances stability, with the cost of a reduction in uniqueness.

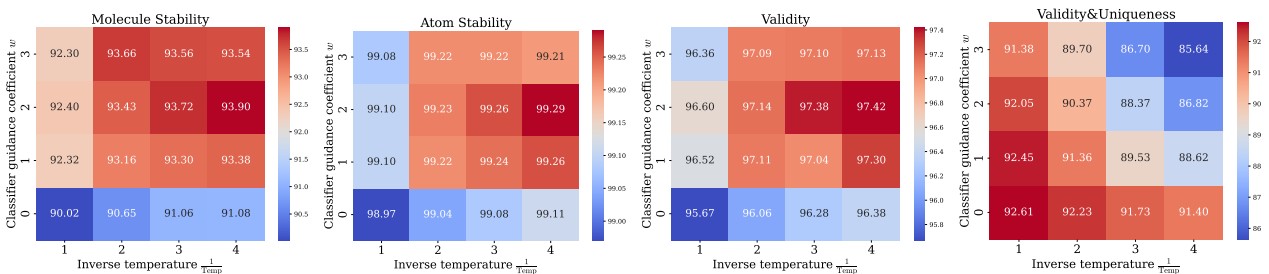

**Fewer-step Sampling of GeoRCG (Semla)**   We present the performance of GeoRCG (Semla) across varying numbers of sampling steps in Table 9. The results demonstrate consistent improvements over SemlaFlow, highlighting the effectiveness of GeoRCG on this advanced method with reduced sampling steps.

**Ablation Study: Representation Encoders**   Geometric representations play a pivotal role in GeoRCG. To evaluate the importance of representation quality, we conduct an ablation study comparing the quality of molecule samples generated by GeoRCG trained under different geometric encoder configurations.

We first assess **the benefits provided by the pre-training stage**. Specifically, we utilize the pre-trained encoder Frad (Feng et al., 2023), trained on the PCQM4Mv2 dataset (Nakata & Shimazaki, 2017) with a hybrid coordinates denoising task (Feng et al., 2023). This approach has been proven to equivalently learn force fields (Feng et al., 2023; Zaidi et al., 2022), and is

Table 9: Comparison between SemlaFlow and GeoRCG (Semla) across varying numbers of sampling steps. Results of SemlaFlow is obtained by our experiments.

| # Steps | Method | Energy ↓ | Strain ↓ | Atom-Stab. ↑ | Mol.-Stab. ↑ | Validity |
|---|---|---|---|---|---|---|
| | | kcal·mol$^{-1}$ | kcal·mol$^{-1}$ | % | % | |
| 100 | SemlaFlow | 95.72(1.24) | 56.42(1.07) | **99.8(0.00)** | 97.4(0.07) | 94.4(0.17) |
| | GeoRCG (Semla) | **88.6(1.03)** | **47.64(1.10)** | **99.8(0.00)** | **97.6(0.00)** | 95.3(0.13) |
| 50 | SemlaFlow | 100.60(0.55) | 60.31(0.13) | **99.8(0.00)** | 96.9(0.11) | 94.6(0.17) |
| | GeoRCG (Semla) | **91.60(1.03)** | **50.50(0.65)** | **99.8(0.00)** | **97.2(0.20)** | **95.3(0.22)** |
| 20 | SemlaFlow | 117.03(1.37) | 76.99(1.18) | **99.7(0.01)** | 95.4(0.17) | 93.2(0.30) |
| | GeoRCG (Semla) | **99.83(0.91)** | **62.88(0.56)** | **99.7(0.00)** | **95.5(0.13)** | **94.2(0.01)** |

therefore expected to produce informative representations that capture high-level molecular information. We train GeoRCG (EDM) using representations from a well-pretrained Frad and a Frad with randomly initialized weights. The molecule generation quality on QM9, as shown in Table 10, clearly **underscores the critical role of pre-training on large datasets with advanced techniques in improving representation quality, ultimately enhancing GeoRCG's performance.**

Table 10: Quality of molecules generated by GeoRCG (EDM) with different encoders trained on the QM9 dataset. "Random" indicates that the weights were initialized randomly without any pre-training.

| Encoder / Metrics | Atom Sta (%) | Mol Sta (%) | Valid (%) | Valid & Unique (%) |
|---|---|---|---|---|
| Random Enc | 98.55(0.01) | 78.66(0.07) | 94.68(0.09) | 55.99(0.83) |
| Pretrained Enc | **99.10(0.02)** | **92.15(0.23)** | **96.48(0.08)** | **92.45(0.21)** |

Next, we investigate **the impact of different pre-trained encoders**, which could vary in model structure and proxy tasks used for pre-training. Specifically, we compare Unimol (Zhou et al., 2023), which employs a message-passing neural network framework incorporating distance features (i.e., DisGNN in (Li et al., 2023; 2024b)) and primarily uses naive coordinates denoising, with Frad (Feng et al., 2023), which adopts TorchMD (Thölke & De Fabritiis, 2022) as the backbone and utilizes carefully designed hybrid-denoising tasks. Both Unimol (Zhou et al., 2023) and Frad (Feng et al., 2023) are pre-trained on the GEOM-DRUG dataset until convergence. We visualize the t-SNE of the representations generated for GEOM-DRUG. As shown in Figure 7, the t-SNE of the Unimol representations exhibits a clearer clustering pattern based on node numbers compared to the Frad representations, which may suggest better representation learning. To further investigate, we utilize both encoders to train GeoRCG and subsequently evaluate the quality of molecule generation. The Frad-based GeoRCG achieves a Validity of $96.9(0.44)$ and Atom Stability of $84.4(0.27)$, while the Unimol-based GeoRCG achieves a Validity of $98.5(0.12)$ and Atom Stability of $84.3(0.12)$: Although the Frad-based GeoRCG produces slightly higher atom stability, its high variance and significantly lower validity suggest inferior performance. These findings, along with our main results, offer insights into the types of representations more effective for guiding molecule generation, suggesting that **sensitivity to molecule size may be a critical factor**.

Table 11: Quality of molecules generated by GeoRCG (EDM) trained on the QM9 dataset, with and without representation perturbation and representation condition dropout.

| Hyper-parameters / Metrics | | Atom Sta (%) | Mol Sta (%) | Valid (%) | Valid & Unique (%) |
|---|---|---|---|---|---|
| rep noise ✗ | cond. dropout ✗ | 98.53(0.08) | 86.93(0.5) | 93.69(0.09) | 89.12(0.21) |
| rep noise ✗ | cond. dropout ✓ | 98.62(0.08) | 87.9(0.35) | 94.64(0.18) | 90.15(0.02) |
| rep noise ✓ | cond. dropout ✗ | 99.05(0.01) | 91.69(0.08) | 96.48(0.11) | 92.38(0.12) |
| rep noise ✓ | cond. dropout ✓ | **99.10(0.02)** | **92.15(0.23)** | **96.48(0.08)** | **92.45(0.21)** |

**Ablation Study: Representation Perturbation** As discussed in Appendix A, we investigate the effectiveness of the straightforward representation perturbation technique by introducing random noise to perturb the representations during training. Additionally, we apply extra dropout in the conditioning module of our molecule generator to mitigate overfitting on the representation conditions. Ablation experiments presented in Table 11 demonstrate the efficacy of these simple yet impactful methods.

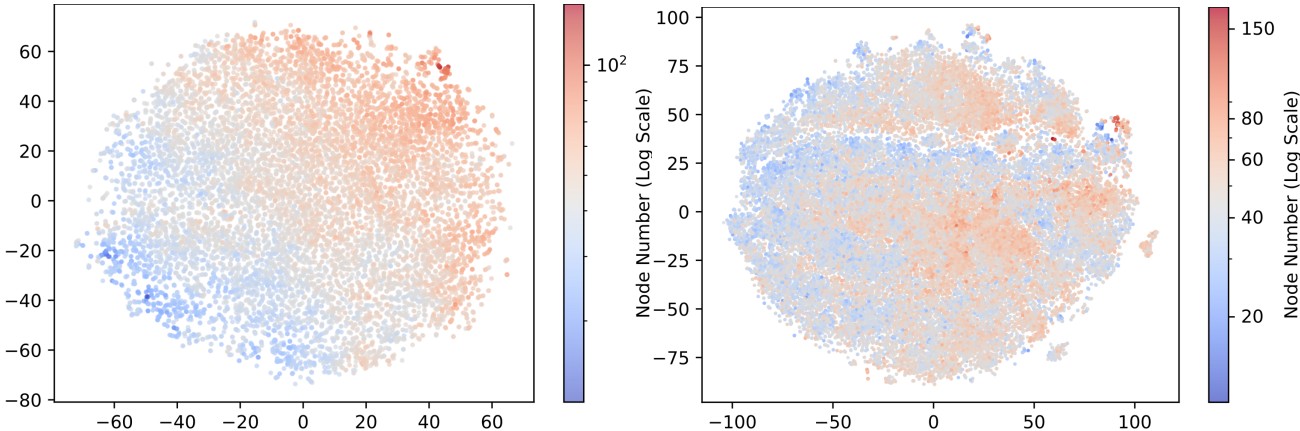

Figure 7: t-SNE visualization of representations produced by the pre-trained encoders for the GEOM-DRUG dataset, colored by node number. The left plot corresponds to Unimol (Zhou et al., 2023), and the right plot corresponds to Frad (Feng et al., 2023).

Table 12: Supplementary evaluation of conditionally generated molecules from GeoRCG (EDM). The right side reports metrics for *unconditionally* generated molecules from other methods for reference. Note that conditional models (left) were trained on half of the QM9 dataset, while unconditional models (right) were trained on the full dataset, which may account for slight decreases in stability and validity metrics.

| | $\alpha$ | $\Delta\varepsilon$ | $\varepsilon_{\text{HOMO}}$ | $\varepsilon_{\text{LUMO}}$ | $\mu$ | $C_v$ | EDM | GeoLDM | EquiFM |
|---|---|---|---|---|---|---|---|---|---|
| Atom Sta (%) | 98.93(0.04) | 98.95(0.03) | 98.93(0.02) | 98.99(0.01) | 98.90(0.02) | 98.98(0.02) | 98.7 | 98.9(0.1) | 98.9(0.1) |
| Mol Sta (%) | 90.64(0.36) | 90.46(0.24) | 90.38(0.19) | 90.94(0.23) | 90.02(0.22) | 90.87(0.12) | 82 | 89.4(0.5) | 88.3(0.3) |
| Valid (%) | 95.40(0.04) | 95.44(0.11) | 95.46(0.19) | 95.74(0.16) | 95.26(0.04) | 95.62(0.08) | 91.9 | 93.8(0.4) | 94.7(0.4) |
| Valid & Unique (%) | 90.47(0.44) | 90.09(0.13) | 90.06(0.11) | 90.32(0.24) | 89.93(0.24) | 90.20(0.16) | 90.7 | 92.7(0.5) | 93.5(0.3) |
| Valid & Unique & Novelty (%) | 50.03(0.26) | 50.81(0.15) | 50.90(0.33) | 50.59(0.08) | 51.70(0.79) | 51.10(0.29) | 65.7 | 58.1 | 57.4 |

**Quality of Conditionally Generated Molecules**    Detailed molecular metrics for conditionally generated molecules are provided in Table 12. For comparison, we also include the stability metrics of molecules conditionally generated by EDM, which highlight a notable improvement in stability with GeoRCG. Specifically, EDM's stability scores are: $\alpha$ (80.4%), $\Delta\varepsilon$ (81.73%), $\varepsilon_{\text{HOMO}}$ (82.81%), $\varepsilon_{\text{LUMO}}$ (83.6%), $\mu$ (83.3%), and $C_v$ (81.03%).

## D. Theoretical Analysis

In this section, we provide rigorous theoretical analysis on representation-conditioned diffusion models. Our theory is not limited to molecule generation, and is the first theoretical breakthrough for the RCG framework (Li et al., 2023).

Our analysis is organized as follows. In Appendix D.1, we analyze the generation bound of representation-conditioned diffusion models in *unconditional generation* tasks by showing: (i) the representation can be well generated by the first-stage diffusion model with mild assumptions (Appendix D.1.1); (ii) the second-stage representation-conditioned diffusion model exhibits no higher generalization error than traditional one-stage diffusion model, and can arguably achieve lower error leveraging the informative representations (Appendix D.1.2). Then in Appendix D.2, we analyze *conditional generation* tasks as follows: (iii) under mild assumptions of representations and targets, we provide novel bound for score estimation error (Appendix D.2.1); (iv) generated representations have provable reward improvement towards the target, with the suboptimality composed of offline regression error and diffusion distribution shift (Appendix D.2.2), thus would improve the second stage of conditional generation (Appendix D.2.3).

**Notations.**    In this section, we use SDE and score matching formulations of diffusion models to present our theoretical results, given their equivalence with the DDPM family (Song et al., 2021). We consider the random variable $x \in \mathbb{R}^{N \times (d+3)}$, and use $q(\cdot)$ to denote the ground truth distributions, $p(\cdot)$ to denote the posterior distribution predicted by diffusion models. For instance, $q(x)$ is the ground truth distribution of the underlying data $x$, while $p_\varphi(r)$ is the predicted distribution of latent representations. We use $T$ to denote the total time of diffusion models, and $N_d$ to represent the discretization step number.

We consider a SDE with continuous time $[0, T]$, as well as its discretized DDPM which has $N_d$ diffusion steps with step size $h := T/N_d$. The forward process is denoted as $(x_t)_{t \in [0,T]} \sim q_t$, and the reverse process is denoted as $(\bar{x}_t)_{t \in [0,T]} \sim p_t$. If the reverse process is predicted by the score matching network, we use its parameters as the subscript. Please note that there are two different initialization of the reverse process: the end of forward process $q_T$ and standard Gaussian noise $\gamma^d$. We use superscript $q_T$ to differentiate the former from the latter.

## D.1. Unconditional Generation

### D.1.1. PROVABLE GENERATION OF REPRESENTATIONS

Recall the two-stage generation process of representation-conditioned generation: $p(x, r) = p_\theta(x|r) p_\varphi(r)$. To quantitatively evaluate the generation process, we consider two stages separately. In this subsection, we first provide theoretical analysis on the provable generation of representations $p_\varphi(r)$.

**Assumption D.1.** (Second moment bound of representations.)

$$m_r^2 := \mathbb{E}_{q(r)}[||r - \bar{r}||^2] < \infty \tag{11}$$

where $q(r)$ is the ground truth distribution of the representations, and $\bar{r} := \mathbb{E}_{q(r)}[r]$.

**Assumption D.2.** (Lipschitz score of representations). For all $t \geq 0$, the score $\nabla \ln q(r_t)$ is $L_r$-Lipschitz.

where $q(r_t)$ is the distribution of noisy latent $r_t$ at diffusion step $t$ in the forward process.

Finally, the quality of diffusion models obviously depends on the expressivity of score network $\varphi$ with prediction $s_\varphi^{(t)}$.

**Assumption D.3.** (Score estimation error of representations). For all $t \in [0, T]$,

$$\mathbb{E}_{q(r_t)}[||s_\varphi^{(t)} - \nabla \ln q(r_t)||^2] \leq \epsilon_{\varphi,\text{score}}^2 \tag{12}$$

These are similar assumptions to the ones in (Chen et al., 2023).

**Proposition D.4.** *Suppose Assumption D.1, Assumption D.2, Assumption D.3 hold, and the step size $h := T/N_d$ satisfies $h \preceq 1/L_r$. Then the following holds,*

$$\text{TV}(p_\varphi(r_0), q(r)) \preceq \underbrace{\sqrt{\text{KL}(q(r)||\gamma^{d_r})} \exp(-T)}_{\text{convergence of forward process}} + \underbrace{(L_r \sqrt{d_r h} + L_r m_r h)\sqrt{T}}_{\text{discretization error}} + \underbrace{\epsilon_{\varphi,\text{score}} \sqrt{T}}_{\text{score estimation error}} \tag{13}$$

Here $a(\cdot) \preceq b(\cdot)$ means there exists a constant $C$ such that $a(\cdot) \leq Cb(\cdot)$ always holds. This is a direct conclusion from (Chen et al., 2023). In typical DDPM implementation, we choose $h = 1$ and thus $T = N_d$. Remarkably, Proposition D.4 indicates the benefit of generating the representation first: since $d_r \ll d$, the generation quality (measured by the TV distance in Proposition D.4) of the low-dimensional representation can easily outperform directly generating the high-dimensional data points $x$. The theorem also accounts for applying a lightweight MLP as the denoising network while in the stage of generating the representation.

### D.1.2. PROVABLE SECOND-STAGE GENERATION

**Tractable Training Loss.** Now we analyze the generation quality of the second-stage diffusion model. Since we sample from $p_\theta(x, r)$, we have representations as conditions even for unconditional generation tasks. To learn the score function conditioning on the representations, consider the following loss for score matching,

$$\mathcal{L}(\theta) = \int_0^T \lambda(t) \mathbb{E}_{x_t, r}[||s_\theta(x_t, r, t) - \nabla_{x_t} \log q_{t|r}(x_t|r)||^2] \mathrm{d}t \tag{14}$$

However, since $q_{t|r}(x_t|r)$ is intractable, we use the following equivalent losses:

$$\mathcal{L}(\theta) = \int_0^T \lambda(t) \mathbb{E}_{x_0, r}\Big[\mathbb{E}_{x_t|x_0}[||s_\theta(x_t, r, t) - \nabla_{x_t} \log q_{t|0}(x_t|x_0)||^2|x_0]\Big] \mathrm{d}t + C \tag{15}$$

**Proposition D.5.** *(Tractable representation-conditioned score matching loss.)*

$$\mathcal{L}(\theta) := \int_0^T \lambda(t)\mathbb{E}_{x_t,r}[||s_\theta(x_t,r,t) - \nabla_{x_t}\log q_{t|r}(x_t|r)||^2]dt \tag{16}$$

$$= \int_0^T \lambda(t)\mathbb{E}_{x_0,r}\Big[\mathbb{E}_{x_t|x_0}\big[||s_\theta(x_t,r,t) - \nabla_{x_t}\log q_{t|0}(x_t|x_0)||^2|x_0\big]\Big]dt + C \tag{17}$$

*Proof.* The key is the following important property holds since the gradient is taken w.r.t. $x_t$ only:

$$\nabla_{x_t}\log q_{t|r}(x_t|r) = \nabla_{x_t}\log q_{t,r}(x_t,r) \tag{18}$$

The remaining of the derivation parallels to traditional DDPM. We can replace $\nabla_{x_t}\log q_{t,r}(x_t,r)$ with $\nabla x_t \log q_{t,r|0}(x_t,r|x_0)$:

$$\nabla x_t \log q_{t,r}(x_t,r) = \mathbb{E}_{x_0,r|x_t}\Big[\nabla_{x_t}\log q_{t,r|0}(x_t,r|x_0)\Big|x_t\Big] \tag{19}$$

Thus,

$$\mathbb{E}_r\mathbb{E}_{x_t\sim q(x_t|r)}[||s_\theta(x_t,r,t) - \nabla_{x_t}\log q_{t|r}(x_t|r)||^2] \tag{20}$$

$$=\mathbb{E}_r\mathbb{E}_{x_0\sim q(x_0|r)}\mathbb{E}_{x_t\sim q(x_t|r,x_0)}[||s_\theta(x_t,r,t) - \nabla_{x_t}\log q_{t|r}(x_t|x_0,r)||^2] \tag{21}$$

$$=\mathbb{E}_r\mathbb{E}_{x_0\sim q(x_0|r)}\mathbb{E}_{x_t\sim q(x_t|x_0)}[||s_\theta(x_t,r,t) - \nabla_{x_t}\log q_{t|r}(x_t|x_0)||^2] \tag{22}$$

which is equivalent to our tractable score matching loss. □

**Rigorous Error Bound for Second-Stage Generation.** Utilizing Proposition D.5, analysis of the second-stage diffusion parallels to the first stage, except that the score network takes additional inputs $r$.

**Assumption D.6.** (Second moment bound of molecule features.)

$$m_x^2 := \mathbb{E}_{q(x)}[||x - \bar{x}||^2] < \infty \tag{23}$$

where $q(x)$ is the ground truth distribution of the molecule features, and $\bar{x} := \mathbb{E}_{q(x)}x$.

**Assumption D.7.** (Lipschitz score of second stage). For all $t \geq 0$, the score $\nabla \ln q(x_t)$ is $L_x$-Lipschitz.

where $q(x_t)$ is the distribution of noisy latent $x_t$ at diffusion step $t$ in the forward process.

Finally, we make some assumptions of the score network estimation error.

**Assumption D.8.** (Score estimation error of second-stage diffusion). For all $t \in [0,T]$,

$$\mathbb{E}_{r\sim p_\varphi(r),x_t\sim q_t(x_t)}[||s_\theta(x_t,t,r) - \nabla \ln q_t(x_t)||^2] \leq \epsilon_{\theta,\text{score}}^2 \tag{24}$$

This assumption contains the error brought by generating representations, i.e., the TV distance shown in Proposition D.4. Later in Theorem D.12 we explicitly deal with the error brought by representation generation, which results in a more fine-grained error bound.

We now present a key lemma which facilitates analysis and the proof of the central Theorem D.12.

**Lemma D.9.** *Suppose Assumption D.6, Assumption D.7, Assumption D.8 hold, and the step size $h := T/N_d$ satisfies $h \preceq 1/L_x$. Suppose we sample $x \sim p_\theta(x|r)$ from Gaussian noise where $r \sim p_\varphi(r)$, and denote the final distribution of $x$ as $p_{\theta,\varphi}(x)$. Then the following holds,*

$$\text{TV}(p_{\theta,\varphi}(x),q(x)) \preceq \underbrace{\sqrt{\text{KL}(q(x)||\gamma^{N(d+3)})}\exp(-T)}_{\textit{convergence of forward process}} + \underbrace{(L_x\sqrt{N(d+3)h} + L_xm_xh)\sqrt{T}}_{\textit{discretization error}} + \underbrace{\epsilon_{\theta,\text{score}}\sqrt{T}}_{\textit{score estimation error}} \tag{25}$$

*Proof.* Recall the notation that $p_{\theta,\varphi}(x) := \int_r p_{0|r}(x_0|r) p_r(r) \mathrm{d}r = p_0$ predicted by denoising networks $\theta, \varphi$ starting from Gaussian noise $\gamma^{N(d+3)}$. Consider the reverse process $p_0^{q_T}(x_0)$ starting from $q_T$ instead of $\gamma^{N(d+3)}$,

$$\mathrm{TV}(p_0, q(x)) \leq \mathrm{TV}(p_0, p_0^{q_T}) + \mathrm{TV}(p_0^{q_T}, q_0) \tag{26}$$

Using the convergence of the OU process in KL divergence (see (Chen et al., 2023)), the following holds for the first term,

$$\mathrm{TV}(p_0, p_0^{q_T}) \leq \mathrm{TV}(\gamma^{N(d+3)}, q_T) \leq \sqrt{\mathrm{KL}(q(x)||\gamma^{N(d+3)})} \exp(-T) \tag{27}$$

The second term is caused by score estimation error and discretization error, which can be bounded by

$$\mathrm{TV}(p_0^{q_T}, q_0)^2 \leq \mathrm{KL}(q_0||p_0^{q_T}) \preceq (\epsilon_{\theta,\mathrm{score}}^2 + L_x^2 N(d+3)h + L_x^2 m_x^2 h^2)T \tag{28}$$

We start proving Equation (28) by proving

$$\sum_{k=1}^{N_d} \mathbb{E}_{q_0, r \sim p_\varphi} \int_{(k-1)h}^{kh} ||s_\theta^{(kh)}(x_{kh}, kh, r) - \nabla \ln q_t(x_t)||^2 \mathrm{d}t \preceq (\epsilon_{\theta,\mathrm{score}}^2 + L_x^2 N(d+3)h + L_x^2 m_x^2 h^2)T \tag{29}$$

For $t \in [(k-1)h, kh]$, we decompose

$$\mathbb{E}_{q_0, r \sim p_\varphi}[||s_\theta^{(kh)}(x_{kh}, kh, r) - \nabla \ln q_t(x_t)||^2] \tag{30}$$

$$\preceq \mathbb{E}_{q_0, r \sim p_\varphi}[||s_\theta^{(kh)}(x_{kh}, kh, r) - \nabla q_{kh}(x_{kh})||^2] + \mathbb{E}_{q_0}[||\nabla q_{kh}(x_{kh}) - \nabla q_t(x_{kh})||^2] \tag{31}$$

$$+ \mathbb{E}_{q_0}[||\nabla q_t(x_{kh}) - \nabla q_t(x_t)||^2] \tag{32}$$

$$\preceq \epsilon_{\theta,\mathrm{score}}^2 + \mathbb{E}_{q_0}\left[\left|\left|\nabla \ln \frac{q_{kh}}{q_t}(x_{kh})\right|\right|^2\right] + L_x^2 \mathbb{E}_{q_0}[||x_{kh} - x_t||^2] \tag{33}$$

Note that we omit the term $r$ in expectation of last two terms because they are independent of $r$.

Utilizing Lemma 16 from (Chen et al., 2023), we bound

$$\left|\left|\nabla \ln \frac{q_{kh}}{q_t}(x_{kh})\right|\right|^2 \preceq L_x^2 N(d+3)h + L_x^2 h^2 ||x_{kh}||^2 + (L_x^2 + 1)h^2 ||\nabla \ln q_t(x_{kh})||^2 \tag{34}$$

For the last term,

$$||\nabla \ln q_t(x_{kh})||^2 \preceq ||\nabla \ln q_t(x_t)||^2 + ||\nabla \ln q_t(x_{kh}) - \nabla \ln q_t(x_t)||^2 \tag{35}$$

$$\preceq ||\nabla \ln q_t(x_t)||^2 + L_x^2 ||x_{kh} - x_t||^2 \tag{36}$$

where the second term is absorbed into the third term of the decomposition Equation (30). Thus,

$$\mathbb{E}_{q_0, r \sim p_\varphi}[||s_\theta^{(kh)}(x_{kh}, kh, r) - \nabla \ln q_t(x_t)||^2] \tag{37}$$

$$\preceq \epsilon_{\theta,\mathrm{score}}^2 + L_x^2 N(d+3)h + L_x^2 h^2 \mathbb{E}_{q_0}[||x_{kh}||^2] + L_x^2 h^2 \mathbb{E}_{q_0}[||\nabla \ln q_t(x_t)||^2] + L_x^2 \mathbb{E}_{q_0}[||x_{kh} - x_t||^2] \tag{38}$$

$$\preceq \epsilon_{\theta,\mathrm{score}}^2 + L_x^2 N(d+3)h + L_x^2 h^2(N(d+3) + m_x^2) + L_x^3 N(d+3)h^2 + L_x^2(m_x^2 h^2 + N(d+3)h) \tag{39}$$

$$\preceq \epsilon_{\theta,\mathrm{score}}^2 + L_x^2 N(d+3)h + L_x^2 h^2 m_x^2 \tag{40}$$

Analogous to (Chen et al., 2023), using properties of Brownian motions and local martingales, we can apply Girsanov's theorem and complete the stochastic integration. Since $q_0$ is the end of the reverse SDE, by the lower semicontinuity of the KL divergence and the data-processing inequality (Beaudry & Renner, 2011), we take the limit and obtain

$$\mathrm{KL}(q_0||p_0^{q_T}) \preceq (\epsilon_{\theta,\mathrm{score}}^2 + L_x^2 N(d+3)h + L_x^2 h^2 m_x^2)T \tag{41}$$

We finally conclude with Pinsker's inequality ($\mathrm{TV}^2 \leq \mathrm{KL}$). $\qquad\square$

This result holds for *general* representation-conditioned diffusion models, and to our best knowledge we are the first to provide theories for representation-conditioned generation, which is a general generation framework suitable for various domains such as images (Li et al., 2023) and graphs.

Lemma D.9 quantitatively characterizes the bound on generalization error in representation-conditioned diffusion. It directly suggests that the error of representation-conditioned diffusion will be no higher than that of its one-stage counterpart. This is because the first two components of the generalization error (i.e., the convergence of the forward process and the discretization error) of the representation-conditioned diffusion model align with those of traditional DDPM, provided that both are parameterized using the same diffusion processes. Furthermore, the third component (score estimation error) can be made identical if we simply set all representation-relevant parameters in $s_\theta$ to zero and disregard representation's impact. We therefore have the following conclusion,

**Corollary D.10.** *Self-representation-conditioned diffusion model can have the same or a lower generation distribution error than one-stage diffusion model.*

We now give a more *fine-grained error bound* analysis of representation-conditioned diffusion, given the relationship between $r$ and $x$ that enables our further qualitative analysis for the argubly better performance.

**Assumption D.11.** (representation-conditioned score estimation error of second-stage diffusion). For all $t \in [0, T]$,

$$\mathbb{E}_{r \sim p_\varphi(r), x_t \sim q_t(x_t|r)}[||s_\theta(x_t, t, r) - \nabla \ln q_t(x_t|r)||^2] \leq \epsilon_{\varphi,\theta,\text{cond}}^2 \tag{42}$$

The following main theorem is novel and precise since it (i) deals with the generation error of first-stage representations explicitly; (ii) takes advantages of the conditional distribution $q(x|r)$ in the denoising network.

**Theorem D.12.** *(Theorem 3.2 in the main text) Suppose Assumption D.6, Assumption D.7, Assumption D.11 hold, and the step size $h := T/N_d$ satisfies $h \preceq 1/L_x$. Suppose we sample $x \sim p_\theta(x|r)$ from Gaussian noise where $r \sim p_\varphi(r)$, and denote the final distribution of $x$ as $p_{\theta,\varphi}(x)$. Define $p_0^{q_T|\varphi}$, which is the end point of the reverse process starting from $q_{T|\varphi}$ instead of Gaussian. Here $q_{T|\varphi}$ is the $T$-th step in the forward process starting from $q_{0|\varphi} := \frac{1}{A} \int_r q(x_0|r)p_\varphi(r)\mathrm{d}r$ where $A$ is the normalization factor. Then the following holds,*

$$\text{TV}(p_{\theta,\varphi}(x), q(x)) \preceq \underbrace{\sqrt{\text{KL}(q_{0|\varphi}||\gamma^{N(d+3)})} \exp(-T)}_{\text{convergence of forward process}} + \underbrace{(L_x\sqrt{N(d+3)h} + L_x m_x h)\sqrt{T}}_{\text{discretization error}} \tag{43}$$

$$+ \underbrace{\epsilon_{\varphi,\theta,\text{cond}}\sqrt{T}}_{\text{conditional score estimation error}} + \underbrace{\text{TV}(q_{0|\varphi}, q_0)}_{\text{representation generation error}} \tag{44}$$

*Proof.* The proof sketch parallels that of Lemma D.9, except that in the first step we decompose the TV distance as follows,

$$\text{TV}(p_{\theta,\phi}, q(x)) \leq \text{TV}(p_0, p_0^{q_T|\varphi}) + \text{TV}(p_0^{q_T|\varphi}, q_{0|\varphi}) + \text{TV}(q_{0|\varphi}, q_0) \tag{45}$$

We complete the proof analogously to the proof of Lemma D.9. $\square$

Remarkably, when $q_{0|\varphi}$, i.e., $p_\varphi$ fully recovers the ground truth marginal distribution of representations $q(r)$, Theorem D.12 has the same format as Lemma D.9 but with $\epsilon_{\varphi,\theta,\text{cond}} < \epsilon_{\theta,\text{score}}$. This is because the former is the score estimation error based on explicit relationship between $x$ and $r$ while the latter learns implicitly. Thus, Theorem D.12 is a much tighter bound for representation-conditioned generation. *To the best of our knowledge, this is the first rigorous theoretical analysis on RCG* (Li et al., 2023). We now provide some qualitative discussions on why representations can arguably lead to better generalization error.

Typically, representations are powerful (and sometimes even complete) as they encode key information about $x$ with potential additional knowledge via pretraining tasks (for example, coordinates denoising for molecules (Zaidi et al., 2022; Feng et al., 2023)). Therefore, it is reasonable to expect that score estimation conditioned on representations can be more accurate (i.e., $\epsilon_{\theta,\text{score}}$ could be significantly smaller than when estimating the score without representation conditioning). If the representations are complete—where a special case would be $r = x$—this would greatly assist in predicting the noise. The same applies when $r$ can be properly transformed back to $x$ by a neural network. More generally, there are intermediate cases where $r$ reflects partial information about $x$ (e.g., a multiset of atoms and bonds), which would still aid in improving prediction.

**Extension to Equivariant Diffusion Models.** The previous conclusions are **generic** and can be applied to general representation-conditioned generation. However, so far we only consider traditional diffusion models without taking into account the permutation $\Pi$ and SE(3) transformation $\Omega$ invariance/equivariance of the diffusion model. We thus extend our theory specifically to equivariant diffusion models that operate on symmetry structures, which is the case for our experiments. Moreover, the previous results assume that both the diffusion process and the denoising model treat atom coordinates $\mathbf{x}$ and atom type features $\mathbf{h}$ identically, while the fact is that (1) the noises of atom coordinates always have zero center of mass (CoM), thus actually lies in a subspace with degree of freedom $3(N-1)$ as opposed to $3N$; (2) in the forward process, $\mathbf{x}_t$ and $\mathbf{h}_t$ are conditional independent give $\mathbf{x}_0$ and $\mathbf{h}_0$, and since the denoising network processes coordinates $\mathbf{x}_t$ and $\mathbf{h}_t$ differently, the score estimation error term can be further decomposed as in Assumption D.13.

**Assumption D.13.** (fine-grained representation-conditioned score estimation error of second-stage diffusion). For all $t \in [0, T]$,

$$\mathbb{E}_{r \sim p_\varphi(r), x_t \sim q_t(x_t|r)}[||s_\theta(x_t, t, r) - \nabla \ln q_t(x_t|r)||^2] \tag{46}$$

$$= \mathbb{E}_{r \sim p_\varphi(r), x_t \sim q_t(x_t|r)}[||s_\theta^{(\mathbf{x})}(x_t, t, r) - \nabla_{\mathbf{x}_t} \ln q_t(\mathbf{x}_t|r)||^2 + ||s_\theta^{(\mathbf{h})}(x_t, t, r) - \nabla_{\mathbf{h}_t} \ln q_t(\mathbf{h}_t|r)||^2] \tag{47}$$

$$\leq (\epsilon_{\varphi,\theta,\text{cond}}^{\mathbf{x}})^2 + (\epsilon_{\varphi,\theta,\text{cond}}^{\mathbf{h}})^2 \tag{48}$$

where $s_\theta^{(\mathbf{x})}(\cdot)$ and $s_\theta^{(\mathbf{h})}(\cdot)$ refer to the predicted score of coordinates and atom type features by the score network, respectively.

Combining all the pieces, we conclude the following result Theorem D.14 for equivariant diffusion models that generate 3D coordinates as described above.

**Theorem D.14.** *Suppose Assumption D.6, Assumption D.7, Assumption D.13 hold, and the step size $h := T/N_d$ satisfies $h \preceq 1/L_x$. Suppose we sample $x \sim p_\theta(x|r)$ from Gaussian noise with zero CoM in the coordinate subspace where $r \sim p_\varphi(r)$, and denote the final distribution of $x$ as $p_{\theta,\varphi}(x)$. Define $p_0^{q_{T|\varphi}}$, which is the end point of the reverse process starting from $q_{T|\varphi}$ instead of Gaussian. Here $q_{T|\varphi}$ is the $T$-th step in the forward process starting from $q_{0|\varphi} := \frac{1}{A} \int_r q(x_0|r) p_\varphi(r) \mathrm{d}r$ where $A$ is the normalization factor. Denote $\gamma_{0-CoM}^{N(d+3)}$ the $N(d+3)$-dimensional Gaussian but with zero center of mass in the $N \times 3$-dimensional subspace for coordinates. Denote $\tilde{p}(\cdot)$ as the distribution after acting permutation group $\Pi$ and SE(3) transformation $\Omega$ on the data from $p(\cdot)$. Then the following holds,*

$$\mathrm{TV}(\tilde{p}_{\theta,\varphi}(x), \tilde{q}(x)) := \alpha(p_{\theta,\varphi}, \Pi, \Omega) \mathrm{TV}(p_{\theta,\varphi}(x), q(x)) \tag{49}$$

$$\preceq \alpha(p_{\theta,\varphi}, \Pi, \Omega) \left( \underbrace{\sqrt{\mathrm{KL}(q_{0|\varphi}||\gamma_{0-CoM}^{N(d+3)})} \exp(-T)}_{\text{convergence of forward process}} + \underbrace{(L_x\sqrt{N(d+3)h} + L_x m_x h)\sqrt{T}}_{\text{discretization error}} \right. \tag{50}$$

$$\left. + \underbrace{(\epsilon_{\varphi,\theta,\text{cond}}^{\mathbf{x}} + \epsilon_{\varphi,\theta,\text{cond}}^{\mathbf{h}})\sqrt{T}}_{\text{conditional score estimation error}} + \underbrace{\mathrm{TV}(q_{0|\varphi}, q_0)}_{\text{representation generation error}} \right) \tag{51}$$

*where $\alpha(\cdot) \in [0, 1]$.*

*Proof.* The proof parallels the proof of Theorem D.12 with three distinct parts.

1. Due to the existence of the permutation group $\Pi$ and the SE(3) group $\Omega$, we need to consider the distribution $\tilde{p}_{\theta,\varphi}(x)$. Note the definition $\alpha(p_{\theta,\varphi}, \Pi, \Omega) := \frac{\mathrm{TV}(\tilde{p}_{\theta,\varphi}(x), \tilde{q}(x))}{\mathrm{TV}(p_{\theta,\varphi}(x), q(x))}$, and by data processing inequality (Beaudry & Renner, 2011), $\alpha(p_{\theta,\varphi}, \Pi, \Omega) \in [0, 1]$. Specifically, when the denoising model $p_{\theta,\varphi}$ is constructed invariant/equivariant to permutations $\Pi$ and SE(3) transformations $\Omega$ (which means the model treats all elements in one equivalent class the same), $\alpha(p_{\theta,\varphi}, \Pi, \Omega)$ reaches the minimum; see (You et al., 2023) for further explanations.

2. Since the Gaussian noises for coordinates are sampled from a subspace with zero center of mean, the prior distribution $\gamma^{N(d+3)}$ in Theorem D.12 should be replaced with $\gamma_{0-CoM}^{N(d+3)}$, the $N(d+3)$-dimensional Gaussian but with zero center of mass in the $N \times 3$-dimensional subspace for coordinates. It is notable that the degree of freedom of $\gamma_{0-CoM}^{N(d+3)}$ is actually $N(d+3) - 3$, and the remaining of the proof still holds (Feng et al., 2024).

3. As explained before, in a properly designed forward process, $\mathbf{x}_t$ and $\mathbf{h}_t$ can be conditional independent give $\mathbf{x}_0$ and $\mathbf{h}_0$. Meanwhile, the denoising network processes coordinates $\mathbf{x}_t$ and $\mathbf{h}_t$ differently, the score estimation error term can be

further decomposed as in Assumption D.13; see (Feng et al., 2024) for more details. Therefore, the term $\epsilon_{\varphi,\theta,\text{cond}}\sqrt{T}$ in Theorem D.12 can be replaced by $\sqrt{(\epsilon_{\varphi,\theta,\text{cond}}^{\mathbf{x}})^2 + (\epsilon_{\varphi,\theta,\text{cond}}^{\mathbf{h}})^2}$, which is further bounded by $(\epsilon_{\varphi,\theta,\text{cond}}^{\mathbf{x}} + \epsilon_{\varphi,\theta,\text{cond}}^{\mathbf{h}})$.

$\square$

In conclusion, we provide a detailed characterization of the generalization error of representation-conditioned diffusion models in this subsection. It is important to note that some parameters in our assumptions, such as Lipschitz scores and estimation errors, are not constants; they are functions of the SDE total time $T$ and the number of diffusion steps $N_d$. Our conclusions also explain why representation-conditioned generation methods remain competitive even when the number of second-stage diffusion steps $N_d$ is decreased for *faster generation*. This is because the score estimation error can remain small even when the number of diffusion steps is reduced, given the guidance from representations. As a result, reducing $N_d$ causes a slower increase in $\epsilon_{\varphi,\theta,\text{cond}}(N_d)$ compared to the score estimation error without representation conditioning, leading to representation-conditioned generative models' strong performance with fewer steps.

### D.2. Conditional Generation

In this subsection, we aim to prove that conditional generation using our representation-conditioned generation have probable reward improvement. While we used $c$ to denote conditions in the main text, we use the notation $y$ instead for the targets or "reward" to keep coordinate with existing literature. Denote $q_a := q(\cdot|y = a)$ as the ground truth conditional distribution, and $\hat{p}_a := p(\cdot|y = a)$ the estimated distribution. Suppose the ground truth distribution satisfies $y := f^*(x, r)$ which can be decomposed as

$$f^*(x,r) = g^*(x_\|, r_\|) + h^*(x_\perp, r_\perp) \tag{52}$$

where we denote $x = x_\|$ when $x \sim q(x)$, $r = r_\|$ when $r \sim q(r)$, and $f^*(x, r) = g^*(x, r)$ when $x \sim q(x), r \sim q(r)$.

To start with, we assume a linear relationship between $r$ and $y$, which is reasonable thanks to the powerful pretrained model (which makes the representations helpful in predicting molecule properties and even complete) if some noises are allowed. In detail, the reward is $f^*(x, r) = \hat{w}^\top r + \xi$ and $\xi \sim \mathcal{N}(0, \nu^2)$. In some cases, we may further make Gaussian assumptions on $r$ (WLOG, $r \sim \mathcal{N}(\mu, \Sigma)$) but is not necessary.

#### D.2.1. PARAMETRIC CONDITIONAL SCORE MATCHING ERROR

First, we give a detailed form of the representation score estimation error presented in Assumption D.1 under the assumptions above.

**Lemma D.15.** *For $\delta \geq 0$, with probability $1 - \delta$, the score estimation error $\epsilon_r \simeq \epsilon_{\varphi,\text{score}}$ is bounded by*

$$\frac{1}{T - t_0} \int_{t_0}^T \mathbb{E}_{(r_t,y) \sim q_t}[||\nabla \log q_t(r_t|y) - \hat{s}_\varphi(r_t, y, t)||_2^2]\mathrm{d}t \leq \epsilon_r^2 = \mathcal{O}\Big(\frac{1}{t_0}\sqrt{\frac{\mathcal{N}(\mathcal{S}, \frac{1}{n})d_r^2 \log \frac{1}{\delta}}{n}}\Big) \tag{53}$$

*where $t_0$ is the early stopping time of the SDE, $n$ is the number of samples, $\mathcal{S}$ is the parametric function class of denoising network, and $\mathcal{N}(\mathcal{S}, \frac{1}{n})$ is the log covering number of $\mathcal{S}$. When $\mathcal{S}$ is linearly parameterized, $\mathcal{N}(\mathcal{S}, \frac{1}{n}) = \mathcal{O}(d_r^2 \log(\frac{d_r n}{\nu^2}))$.*

*Proof.* This is a direct extension of Lemma C.1 from (Yuan et al., 2023). Note that we consider the special case where the low-dimensional subspace is the original space (i.e., $A = I_{d_r}$ and $d = D = d_r$ in their paper), and our noised linear assumption between $r$ and $y$ is identical to their pseudo labeling setting (i.e., $\hat{y} = \hat{w}^\top r + \xi$ where $\xi \sim \mathcal{N}(0, \nu^2)$). We only provide the proof sketch here.

When $r$ follows the Gaussian design, some algebra gives

$$\nabla_r \log q_t(r, y) = \frac{\alpha(t)}{h(t)} B_t\Big(\alpha(t)r + \frac{h(t)}{\nu^2}yw\Big) - \frac{1}{h(t)}r \tag{54}$$

where $\alpha(t) = \exp(-t/2), h(t) = 1 - \exp(-t), B(t) = \Big(\alpha^2(t)I_{d_r} + \frac{h(t)}{\nu^2}ww^\top + h(t)\Sigma^{-1}\Big)^{-1}$. We then bound the estimation error with PAC-learning concentration argument by using Dudley's entropy integral to bound the Rademacher

complexity, and obtain

$$\epsilon_r^2 = \mathcal{O}\Big(\frac{1}{t_0}\sqrt{\frac{\mathcal{N}(\mathcal{S},\frac{1}{n})d_r^2\log\frac{1}{\delta}}{n}}\Big) \tag{55}$$

Further, the log covering number of $\mathcal{S}$ under Gaussian design satisfies

$$\mathcal{N}(\mathcal{S},\frac{1}{n}) \leq d_r^2\log\Big(1+\frac{d_r n}{t_0\lambda_{\min}\nu^2}\Big) \tag{56}$$

where $0 < \lambda_{\min} < 1$ is the smallest eigenvalue of $\Sigma$, and typically the early stopping time $t_0 = \mathcal{O}(1)$. In (Yuan et al., 2023) the authors assume $\nu^2 = 1/d_r$ which states that the variance $\nu^2$ of regression residuals $\xi$ reduces when we increase the representation dimensions, which is reasonable. $\qquad\square$

Lemma D.15 provides a detailed score estimation error given the linear assumption between $r$ and $y$, which serves as a special case of $\epsilon_{\varphi,\mathrm{score}}^2$. Substituting it into Proposition D.4, we can obtain a quantitative result of representation generation error.

### D.2.2. REWARD IMPROVEMENT VIA CONDITIONAL GENERATION

Next, we want to obtain the reward guarantees of the generated samples give the condition $y$. Define the suboptimality of distribution $P$ as

$$\mathrm{SubOpt}(P;y^*) = y^* - \mathbb{E}_{(x,r)\sim P}[f^*(x,r)] \tag{57}$$

where $y^*$ is the target reward value (condition) and $f^*$ is the ground truth reward function. We use the notation $\hat{p}_a := p_\varphi(r|y^* = a)$, then we have the following result for $\mathrm{SubOpt}(\hat{p}_a; y^* = a)$, which can also be viewed as a form of off-policy regret.

**Lemma D.16.** (Theorem 4.6 in (Yuan et al., 2023).)

$$\mathrm{SubOpt}(\hat{p}_a;y^*=a) \leq \underbrace{\mathbb{E}_{r\sim q_a}\Big[\big|(\hat{w}-w)^\top r\big|\Big]}_{\mathcal{E}_1} + \underbrace{\Big|\mathbb{E}_{r\sim q_a}[g^*(r_\|)] - \mathbb{E}_{r\sim\hat{p}_a}[g^*(r_\|)]\Big|}_{\mathcal{E}_2} + \underbrace{\mathbb{E}_{r\sim\hat{p}_a}[h^*(r_\perp)]}_{\mathcal{E}_3} \tag{58}$$

*Proof.* Recall the notation $q_a := q(\cdot|y = a)$, we have

$$\mathbb{E}_{r\sim\hat{p}_a}[f^*(r)] \tag{59}$$

$$\geq \mathbb{E}_{r\sim q_a}[f^*(r)] - \big|\mathbb{E}_{r\sim\hat{p}_a}[f^*(r)] - \mathbb{E}_{r\sim q_a}[f^*(r)]\big| \tag{60}$$

$$\geq \mathbb{E}_{r\sim q_a}[\hat{f}(r)] - \mathbb{E}_{r\sim q_a}\big[\big|\hat{f}(r) - f^*(r)\big|\big] - \big|\mathbb{E}_{r\sim\hat{p}_a}[f^*(r)] - \mathbb{E}_{r\sim q_a}[f^*(r)]\big| \tag{61}$$

$$\geq \mathbb{E}_{r\sim q_a}[\hat{f}(r)] - \underbrace{\mathbb{E}_{r\sim q_a}\big[\big|\hat{f}(r) - g^*(r)\big|\big]}_{\mathcal{E}_1} - \underbrace{\Big|\mathbb{E}_{r\sim q_a}[g^*(r_\|)] - \mathbb{E}_{r\sim\hat{p}_a}[g^*(r_\|)]\Big|}_{\mathcal{E}_2} - \underbrace{\mathbb{E}_{r\sim\hat{p}_a}[h^*(r_\perp)]}_{\mathcal{E}_3} \tag{62}$$

where $\hat{w}$ is the estimated $w$ by Ridge regression, $\mathbb{E}_{r\sim q_a}[\hat{f}(r)] = \mathbb{E}_{a\sim q}[a]$ and $r = r_\perp, f*(r) = g^*(r)$ when $r \sim q_a$. $\qquad\square$

Here we give a brief interpretation of the decomposition. $\mathcal{E}_1$ is the prediction and generalization error coming from regression, which is independent from the diffusion error. Both $\mathcal{E}_2$ and $\mathcal{E}_3$ come from the diffusion process, where $\mathcal{E}_2$ reflects the disparity between $\hat{p}_a$ and $q_a$ on the subspace support while $\mathcal{E}_3$ measures the off-subspace error in $\hat{p}_a$. The following results give concrete bounds for the terms in Lemma D.16.

**Bounding Regression Error with Offline Bandits.** By estimating $w$ with Ridge regression, we have

$$\hat{w} = (R^\top R + \lambda I)^{-1} R^\top (Rw^* + \eta) \tag{63}$$

where $R^\top = (r_1,\ldots,r_n)$ and $\eta = (\xi_1,\ldots,\xi_n)$ where $\xi_i \sim \mathcal{N}(0,\nu^2)$. Define $V_\lambda := R^\top R + \lambda I$, $\hat{\Sigma}_\lambda := \frac{1}{n}V_\lambda$ and $\Sigma_{q_a} := \mathbb{E}_{r\sim q_a} rr^\top$ and take $\lambda = 1$, we have

**Proposition D.17.** *With high probability,*

$$\mathcal{E}_1 \leq \sqrt{\text{Tr}(\hat{\Sigma}_\lambda^{-1} \Sigma_{q_a})} \cdot \frac{\mathcal{O}(||w^*|| + \nu^2 \sqrt{d_r \log n})}{\sqrt{n}} \tag{64}$$

*Further when $r$ has Gaussian design $r \sim \mathcal{N}(\mu, \Sigma)$,*

$$\text{Tr}(\hat{\Sigma}_\lambda^{-1} \Sigma_{q_a}) \leq \mathcal{O}\big(\frac{a^2}{||w^*||_\Sigma} + d_r\big) \tag{65}$$

*when $n = \Omega(\max\{\frac{1}{\lambda_{\min}}, \frac{d_r}{||w^*||_\Sigma^2}\})$.*

*Proof.* First we have

$$\mathcal{E}_1 = \mathbb{E}_{\hat{p}_a} |r^\top (w^* - \hat{w})| \leq \mathbb{E}_{\hat{p}_a} ||r||_{V_\lambda^{-1}} \cdot ||w^* - \hat{w}||_{V_\lambda} \tag{66}$$

where

$$\mathbb{E}_{\hat{p}_a} ||r||_{V_\lambda^{-1}} \leq \sqrt{\mathbb{E}_{\hat{p}_a} r^\top V_\lambda^{-1} r} = \sqrt{\text{Tr}(V_\lambda^{-1} \mathbb{E}_{\hat{p}_a} r r^\top)} \simeq \sqrt{\text{Tr}(V_\lambda^{-1} \Sigma_{q_a})} \tag{67}$$

Hence we only need to prove

$$||w^* - \hat{w}||_{V_\lambda} \leq \mathcal{O}(||w^*|| + \nu^2 \sqrt{d_r \log n}) \tag{68}$$

Using the closed form expression, we have

$$\hat{w} - w^* = V_\lambda^{-1} R^\top \eta - \lambda V_\lambda^{-1} w^* \tag{69}$$

Thus,

$$||w^* - \hat{w}||_{V_\lambda} \leq ||R^\top \eta||_{V_\lambda^{-1}} + \lambda ||w^*||_{V_\lambda^{-1}} \tag{70}$$

where $\lambda ||w^*||_{V_\lambda^{-1}} \leq \sqrt{\lambda} ||w^*||$, and according to (Abbasi-yadkori et al., 2011),

$$||R^\top \eta||_{V_\lambda^{-1}} = ||R^\top \eta||_{(R^\top R + \lambda I)^{-1}} \leq \mathcal{O}(\nu^2 \sqrt{d_r \log n}) \tag{71}$$

with high probability. We hence conclude the first part of proof.

Further, when $r$ has Gaussian design $r \sim \mathcal{N}(\mu, \Sigma)$, according to Lemma C.6 of (Yuan et al., 2023), we can prove the results. The key here is the conditional distribution follows the Gaussian below,

$$P_r\big(r|\hat{f}(r) = a\big) = \mathcal{N}\Big(\mu + \Sigma\hat{w}(\hat{w}^\top \Sigma\hat{w} + \nu^2)(a - \hat{w}^\top \mu), \Sigma - \Sigma\hat{w}(\hat{w}^\top \Sigma\hat{w} + \nu^2)^{-1}\hat{w}^\top \Sigma\Big) \tag{72}$$

Thus

$$\text{trace}(\hat{\Sigma}_\lambda^{-1} \Sigma_{q_a}) = \text{trace}\left(\frac{\Sigma^{1/2}\hat{w}\hat{w}^\top \Sigma\hat{\Sigma}_\lambda^{-1}\Sigma^{1/2}a^2}{(||\hat{w}||_\Sigma^2 + \nu^2)^2}\right) \leq (1 + \frac{1}{\sqrt{\lambda_{\min}n}}) \cdot \mathcal{O}(\frac{a^2}{||\hat{w}||_\Sigma^2} + d_r) \tag{73}$$

Notice that $||\hat{w}||_\Sigma \geq ||w^*||_\Sigma - ||\hat{w} - w^*||_\Sigma$. We have

$$||\hat{w} - w^*||_\Sigma = \mathcal{O}\left(\frac{||w^*|| + \nu^2 \sqrt{d_r \log(n)}}{\sqrt{n}}\right) \tag{74}$$

where we can prove $||\hat{w}||_\Sigma \geq \frac{1}{2}||w^*||_\Sigma$ when $n = \Omega(\frac{d_r}{||w^*||_\Sigma^2})$. $\qquad\square$

Remarkably, this is a more precise bound improving the results (Lemma C.5 and C.6) in (Yuan et al., 2023), where we make less assumptions on the relationship between $y$ and $r$, explicitly taking $||w||$ and $\nu^2$ into account.

**Bounding Distribution Shift in Diffusion.** We define the distribution shift between two arbitrary distributions $p_1$ and $p_2$ restricted under function class $\mathcal{L}$ as

$$\mathcal{T}(p_1, p_2; \mathcal{L}) := \sup_{l \in \mathcal{L}} \frac{\mathbb{E}_{x \sim p_1}[l(x)]}{\mathbb{E}_{x \sim p_2}[l(x)]} \tag{75}$$

We have the follow lemma.

**Lemma D.18.** *Under the assumption that $r$ follows Gaussian design, then*

$$\mathrm{TV}(\hat{p}_a, q_a) = \mathcal{O}\left( \sqrt{\frac{\mathcal{T}(q(r, y = a), q_{ry}; \mathcal{S})}{\lambda_{\min}}} \cdot \epsilon_r \right) \tag{76}$$

*where $\epsilon_r$ is defined in Lemma D.15. We can bound $\mathcal{E}_2$ with:*

$$\mathcal{E}_2 = \mathcal{O}\left( (\mathrm{TV}(\hat{p}_a, q_a) + t_0)\sqrt{M(a)} \right) \tag{77}$$

*where $M(a) = \mathcal{O}(\frac{a^2}{||w^*||_\Sigma} + d)$. By plugging in $\epsilon_r^2 = \tilde{\mathcal{O}}(\frac{d_r^2}{t_0\sqrt{n}})$, when $t_0 = (d_r^4/n)^{1/6}$, $\mathcal{E}_2$ admits the best trade off with bound*

$$\mathcal{E}_2 = \tilde{\mathcal{O}}\left( \sqrt{\frac{\mathcal{T}(q(r, y = a), q_{ry}; \mathcal{S})}{\lambda_{\min}}} \cdot (d_r^4/n)^{1/6} a \right) \tag{78}$$

*Proof.* The proof directly follows from Lemma C.4 and Lemma C.7 in (Yuan et al., 2023). However, the conclusion is slightly different as we do not assume a low dimensional subspace of $r$. $\qquad\square$

One can also verify that when $r$ follows Gaussian design, $T(q(r, y = a), q_{ry}; \mathcal{S}) = \mathcal{O}(a^2 \vee d_r)$.

### D.2.3. SECOND STAGE OF CONDITIONAL GENERATION

Now that we have proved that: (i) the first-stage diffusion model can estimate the score function with provable error bound (Appendix D.2.1); and (ii) the generated representations have provable reward improvement (Appendix D.2.2). We continue to show that the ultimate generated samples also have distribution shift towards the desired target in the following contexts. Particularly, we want to answer the question: why utilizing the conditionally generated representations is enough for the second stage generation?

First, when we use the generated representations as the only condition of the second stage diffusion model, the generation process is identical to the second stage of unconditional generation. Therefore, the results in Appendix D.1.2 can be directly applied to analyze the second stage generation of conditional generation, which states that the generation conditioning on representations has small TV distance error compared with ground truth conditional distribution. Thus, when we have high-quality first-stage generation, the corresponding second stage generation would introduce almost no additional error, which implies provable reward improvements towards the desired target. In addition, the well-pretrained encode ensures that the correspondence between representations and data points is good, which makes it possible to rigorously construct the data points given the representations (a special case would be $r$ is the complete representation of $x$).

We then partially answer this question from the information theoretic perspectives. We use $H(\cdot)$ to denote the information entropy, and $I(\cdot; \cdot)$ to denote the mutual information between two variables.

- $I(x; r) \geq I(x; y)$. On the one hand, $r$ contains enough information to recover the targets $y$ thanks to the results in Appendix D.2.2, thus we do not explicitly need $y$ for the second stage. One the other hand, benefit from the pretraining task, the representations obviously contains more information in addition to $y$. This assumption is valid if $w^*$ in previous analysis is sparse (there are components in $r$ independent of $y$), i.e., $H(r) > H(y)$. Therefore, generating $x$ conditioning on $r$ is much easier than generating conditioning on $y$ (traditional one stage generation), as the score estimation error of the former one would obviously be much smaller than the latter.

- $I(x, r; y) \geq I(x; y)$. Recall Equation (52) which states the target property $y$ depends on both $x$ and $r$. Hence, $r$ may contain additional information of $y$ obtained from pretrained tasks that is hard to (or cannot) be directly extracted from $x$ - the complex pretrained model assists in extracting relevant information in our two-stage generation, while one-stage generation solely relies on the single denoising model to do so. Therefore, by leveraging representations with provable error bounds, we can better shift the distribution towards the target.

In summary, $r$ is an ideal middle state connecting $x$ and $y$ - it is easy to recover $r$ from $y$ (Appendix D.2.1) and to recover $x$ from $y$ (Appendix D.1.2), and vice versa. In comparison, it is somewhat more difficult to directly recover $x$ from $y$ or predict $y$ from $x$. Consequently, $r$ may be a better indicator of $y$ compared with $x$ due to the aforementioned reasons.

Indeed, one-stage diffusion models generate $x$ directly from conditions $y$ need to optimize a highly complex score $\nabla_x \log p(x|y)$ where $x$ and $y$ are highly non-linearly mapped. As Theorem E.4 in (Yuan et al., 2023) points out, the nonparametric SubOpt of $x$ generated by deep neural networks is larger than our results in Appendix D.2.2, which further validates the advantage of first generating $r$ that can be well mapped from $y$.

## E. Visualization

### E.1. Representation Visualization

To illustrate how well $p_\varphi(r)$ fits $q(r)$, we sample from both $q(r)$ (i.e., the representations produced by the pre-trained encoder for the QM9 and GEOM-DRUG datasets) and the trained representation generator $p_\varphi(r)$. We then visualize them in Appendix E.1, with colors indicating whether the samples are from $q(r)$ or $p_\varphi(r)$. We compute the Silhouette Score of the clustering results, scaled by $10^2$ for clarity. A score close to zero suggests that the two clusters are difficult to distinguish, indicating a good fit between $p_\varphi(r)$ and $q(r)$. Similarly, we provide the visualization of conditionally generated representations in Figure 9

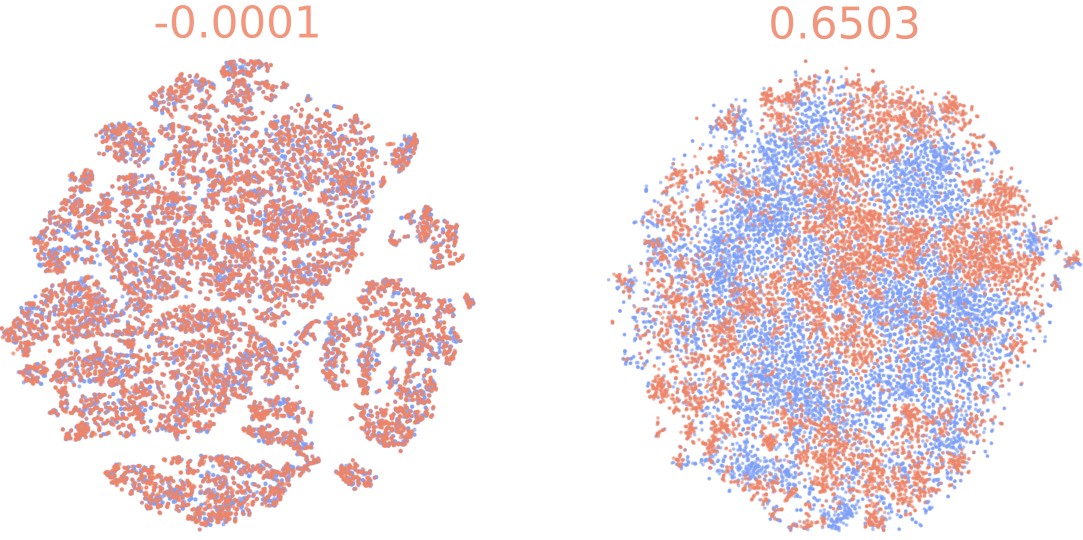

Figure 8: t-SNE visualizations of representations unconditionally generated by the representation generator ($\mathcal{T} = 1.0$) vs. those produced by the pre-trained encoder on the QM9 and GEOM-DRUG datasets. The Silhouette Score is scaled by $10^2$ for clarity.

### E.2. Visualization of Molecule Samples

In this section, we provide additional random molecule samples to offer deeper insights into the performance of GeoRCG. Figure 10 and Figure 11 show unconditional random samples generated by GeoRCG trained on the QM9 and GEOM-DRUG datasets, respectively. Figure 12 presents random samples conditioned on the $\alpha$ property, along with their corresponding errors.

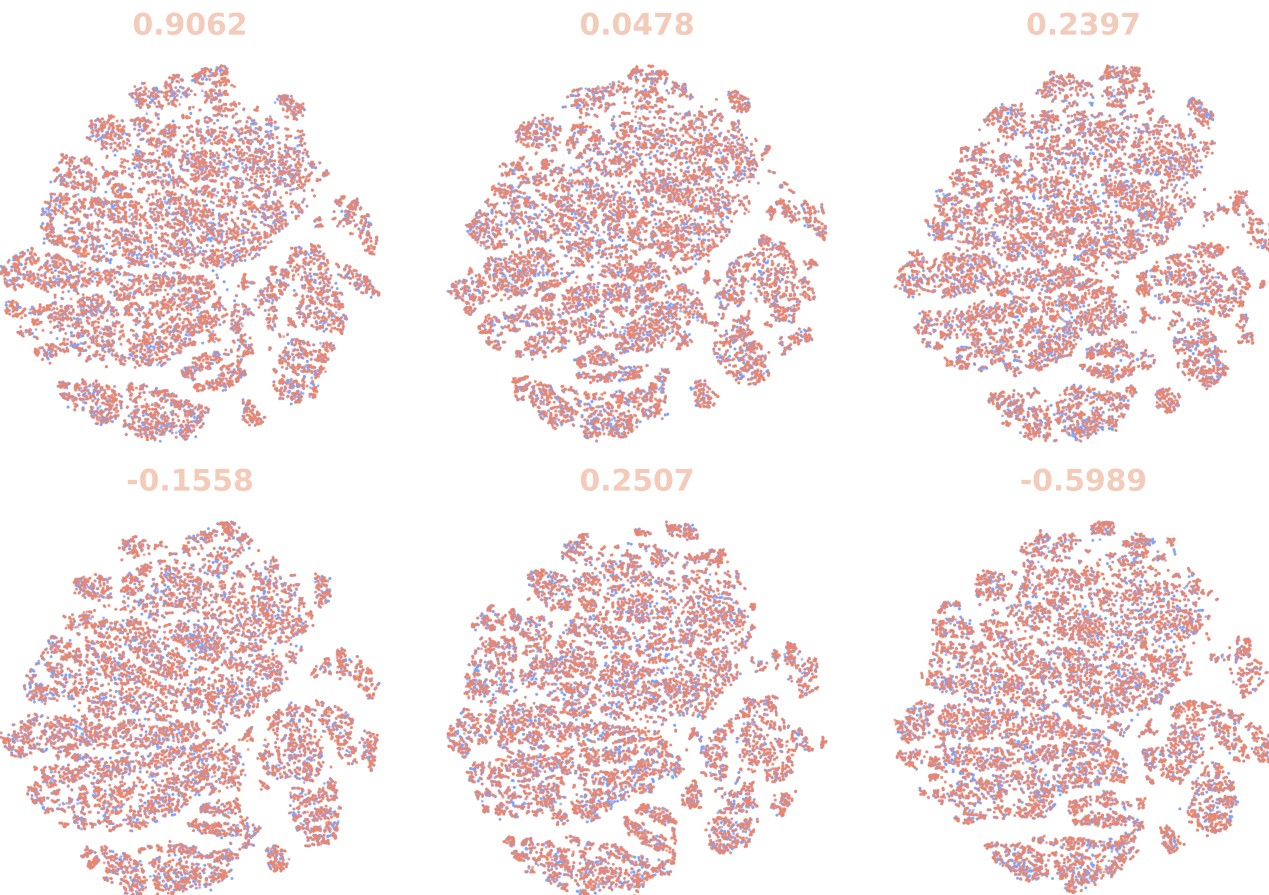

Figure 9: t-SNE visualization of representations conditionally generated by the representation generator vs. those produced by the pre-trained encoder on the QM9 dataset: (a) $\alpha$, (b) $\Delta\epsilon$, (c) $\epsilon_{\text{HOMO}}$, (d) $\epsilon_{\text{LUMO}}$, (e) $\mu$, and (f) $C_v$. The Silhouette Score is scaled by $10^2$ for clarity.

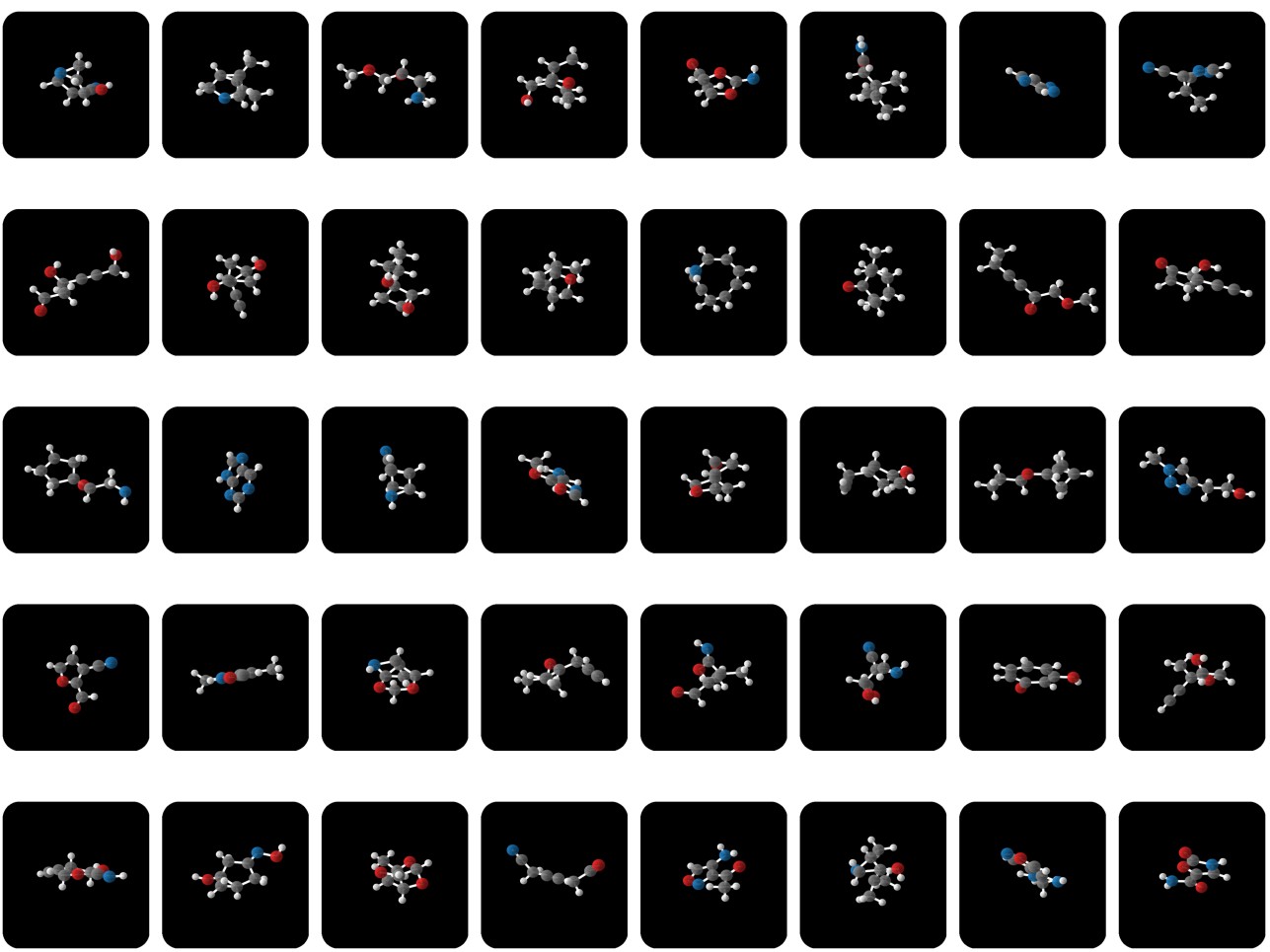

Figure 10: Unconditional random samples from GeoRCG trained on QM9. The number of nodes is randomly sampled from the node distribution $q(N)$.

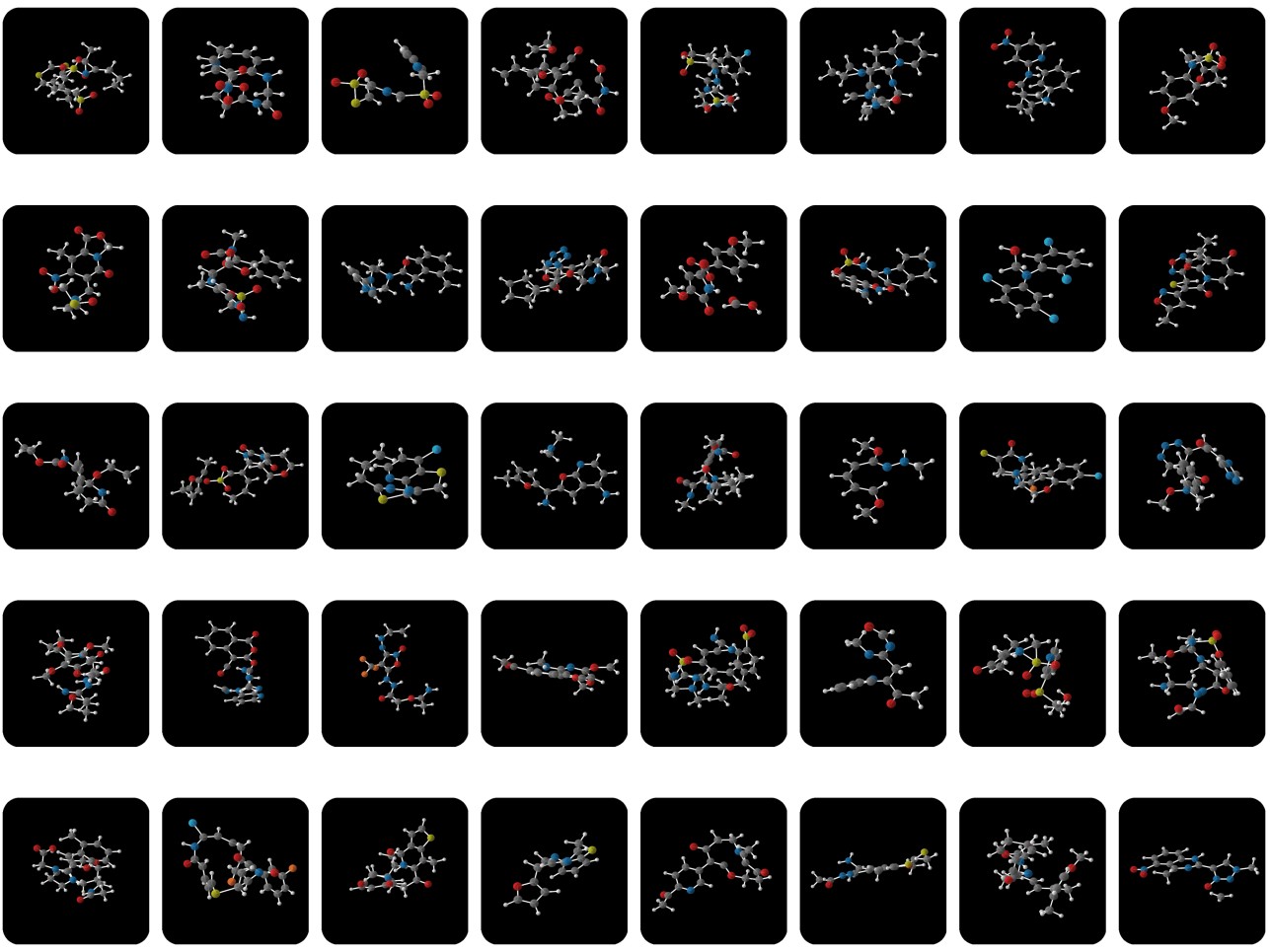

Figure 11: Unconditional random samples from GeoRCG trained on GEOM-DRUG. The number of nodes is randomly sampled from the node distribution $q(N)$.

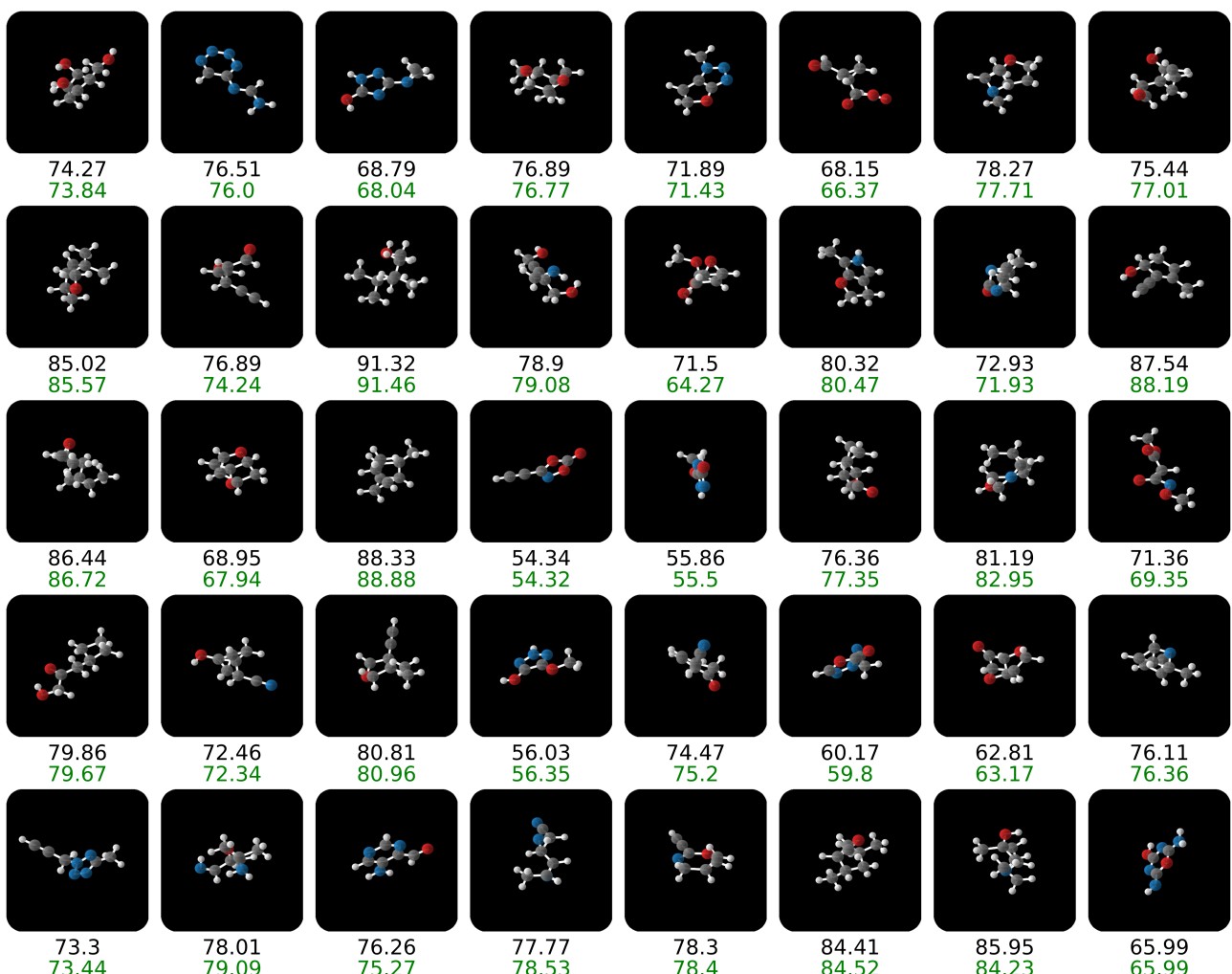

Figure 12: Conditional random samples from GeoRCG trained on QM9 dataset and $\alpha$ property. Black numbers indicate the specified property value condition, while green numbers represent the evaluated property value of the generated samples. The number of nodes and property value conditions are randomly sampled from the joint distribution $q(N, c)$.

