# OpenReview forum: "Geometric Representation Condition Improves Equivariant Molecule Generation"
_ICML.cc/2025/Conference — ICML 2025 spotlightposter_

### Official Review · Reviewer_Tpcv · 2025-03-12

**Overall Recommendation:** 3

**Summary:**

This paper proposes a general framework, namely GeoRCG, for 3D molecule generation. Basically, it factorizes a 3D molecule generation into two stages: the first is to generate a geometric representation and the second step is to generate 3D molecules conditioned on the geometric representation. Such factorization can make the task easier and the resulting model has been shown to achieve great performance on both unconditional and conditional generation.

**Claims And Evidence:**

The main claim is that by the above factorization, the approach can achieve more effectiveness and faster generation. All these claims have been supported by experimental results.

**Essential References Not Discussed:**

N/A

**Experimental Designs Or Analyses:**

I checked all the experimental designs, including unconditional and conditional generation, and they are valid to me.

**Methods And Evaluation Criteria:**

The main method contributions are:

* The novel idea of decomposing molecule generation into two substeps and each of these two steps make sense.
* The two generators, including representation generator and molecule generator, can be trained in parallel.
* The framework can work on both unconditional and conditional generation. While on different conditions, only the representation generator needs to be retrained. This sounds exciting.

The evaluation is performed on widely used benchmarks for molecule generations. The metrics looks valid to me.

**Other Comments Or Suggestions:**

N/A

**Other Strengths And Weaknesses:**

I have included the strengths before. One remaining concern is that since GeoRCG used additional data for the pre-trained encoder, the comparison to other existing molecule generation models might not be strictly fair as they didn't use any data for the pre-trained encoder.

**Questions For Authors:**

Why do you use different pre-trained encoders for QM9 and GEOM-DRUG benchmarking experiments? Does a universal encoder work for both benchmarks?

**Relation To Broader Scientific Literature:**

The proposed two-stage generation strategy are quite novel to the molecule generation field. It brings insights and important performance gains. Even though such two-stage idea has been explored in other domain such as image generation, I believe extending this to 3D molecule generation should be recognized.

**Theoretical Claims:**

I haven’t checked the details of the proof for Theorem 3.2 but the resulting takeaway sounds reasonable to me.

---

> ### Author Rebuttal · Authors · 2025-04-01
>
> Dear Reviewer Tpcv,
>
> We sincerely appreciate your insightful and thorough review, and we are grateful for your recognition of our work. Below, we provide detailed responses to each of your comments.
>
> ---
>
> ### **1. Does additional pretraining data in the representation encoder introduce unfairness?**
>
> Thank you for raising this important point! We believe this question is crucial and warrants further clarification. We argue that **additional pretraining data does not introduce unfairness** for the following reasons:
> + It’s not that “other methods didn’t use any data for the pre-trained encoder,” but rather that they **cannot leverage valuable prior knowledge from additional data** due to the absence of an effective representation guidance mechanism. In contrast, GeoRCG successfully utilizes pre-trained encoders and advanced pre-training techniques for generation purposes. This approach is consistent with practices in the CV domain [1].
> + While it may seem that we use additional pretraining data, generally speaking, **no customized pre-training** or **additional training beyond generative training** is necessary for GeoRCG’s development. We simply leverage **public** pre-trained models, like Frad, that have been trained using **general** pre-training methods! Although we did pre-train models for the DRUG dataset, this was due to the lack of public checkpoints containing rare elements (e.g., Bi in some drugs). However, we believe that as molecular pre-training progresses and the effectiveness of our work, more advanced pre-trained checkpoints will become publicly available, eliminating the need for pre-training ourselves, similar to the CV domain [1, 2].
> + Even when seriously considering the effect of additional data in generative training, we emphasize that the extra training data **solely contributes to forming a meaningful and structured latent representation manifold in the encoder**. During GeoRCG’s training, we still focus exclusively on generative data (i.e., QM9 and DRUG), with the encoder simply translating molecules from this data into structured, meaningful representations.
>
> ### **2. Yes, a universal encoder works for both benchmarks!**
> + We use Frad for QM9 experiments and UniMol for GEOM-DRUG experiments simply **to achieve optimal performance.**
> + When **using Frad for both benchmarks**, GeoRCG **consistently shows improved performance:** As shown in lines 929-943, Frad-based GeoRCG achieves an Atom Stability of 84.4 (compared to EDM’s 81.3) and Validity of 96.9 (compared to EDM’s 92.6). While these results are strong, UniMol-based GeoRCG achieves slightly higher Validity (98.5) with similar Atom Stability (84.3), which is why we selected it as our primary generator.
> + Our analysis in Figure 6 and lines 942-943 further demonstrates that pre-trained encoders with more structured representation distributions tend to yield superior performance.
>
> ---
>
> Once again, we deeply appreciate your valuable feedback and look forward to continued discussions!
>
> [1] Li, Tianhong, Dina Katabi, and Kaiming He. “Return of unconditional generation: A self-supervised representation generation method.”
>
> [2] Li, Tianhong, et al. "Mage: Masked generative encoder to unify representation learning and image synthesis."

---

### Official Review · Reviewer_zxks · 2025-03-12

**Overall Recommendation:** 3

**Summary:**

This paper introduces the GeoRCG framework which is a method for generating molecules by firstly generating a conditioning vector and then generating a molecule based on this condition. The authors show how the previously introduced RCG framework can be applied to molecule generation in 3D space by training a diffusion model to generate a vector which matches the molecular encoding provided by a pretrained geometric molecule embedding model. A 3D molecule generation model can then be trained in parallel to generate a molecule based given the embedding as conditioning vector. This representation makes it simpler to apply a pretrained 3D generative model to conditional generation tasks since only the conditioning vector generative model needs to be updated. For both diffusion and flow-matching molecule generation models, GeoRCG shows improvements in generative performance and shows strong performance on conditional generation tasks.

**Claims And Evidence:**

Yes, the authors claims are well supported by their extensive evaluations.

**Essential References Not Discussed:**

NA

**Experimental Designs Or Analyses:**

The experiments seem to be designed correctly and consistently with existing methods.

**Methods And Evaluation Criteria:**

The evaluation has mostly been performed in a way that is fair and consistent with existing models, although GeoRCG uses a number of additional strategies to improve performance such as low temperature sampling and classifier-free guidance that many existing methods do not use. Discussed further below.

**Other Comments Or Suggestions:**

- Equation 6 should use a subscript on the first $q$ term.

**Other Strengths And Weaknesses:**

- In addition to conditioning on a representation, the authors introduce a number of additional tricks into the model, including low temperature sampling, classifier-free guidance and representation perturbation. It would be better to see how GeoRCG performs without these tricks and show a full ablation table with various combinations of training strategies and compare to existing methods. Figure 5 is a good start but it is important to see how these results compare against existing methods in an otherwise identical setup. This is especially important since one of the main comparisons in the paper is to the existing EDM model since the setup is otherwise the same, but it is difficult to tell how useful the RCG method is if GeoRCG uses other sampling tricks that EDM does not.

**Questions For Authors:**

- What dataset was the UniMol pretrained encoder trained on for the GEOM drugs model, in addition to GEOM drugs?

**Relation To Broader Scientific Literature:**

- The GeoRCG method takes a lot of inspiration from the existing RCG method, but the authors extend this to 3D equivariant generation models and introduce an error bound for the representation-conditioned generative model.

- SemlaFlow focuses a lot of its evaluation on the efficiency of 3D molecule generation, but no evaluation was performed with GeoRCG to check how the new method impacts the inference time. How much overhead is there for generating a representation vector, and for adapting the SemlaFlow model to allow conditioning on this vector?

- The improvement over existing methods for unconditional generation is very small, although the improvement over other methods in the conditional setting is more significant.

**Theoretical Claims:**

The theoretical results in the main appear to be correct although I did not check them thoroughly.

---

> ### Author Rebuttal · Authors · 2025-04-01
>
> Dear Reviewer zxks:
>
> We sincerely appreciate your expert and detailed review! Below, we address each of your comments in detail. **We have provided additional tables in an anonymous GitHub repository: https://anonymous.4open.science/r/rebuttal-8746. (Alternative link in case the previous one encounters issues: https://docs.google.com/document/d/e/2PACX-1vQFD87aQep2Q11albXeuuTKUr9HrrLIALteNrVVsbguQ92c_8ArXr_H43J8xv0WrnuRxbzwAOgBYsav/pub)**
>
> ---
>
> ### **1. Regarding the “very small improvements for unconditional generation”**
> Thank you for the thoughtful comment on this! However, we would like to respectfully clarify that, when we refer to improvements, we are emphasizing the benefits of GeoRCG **over its base model, since we aim to improve any models’ performance.** In this context, **the improvement in unconditional settings is substantial**: GeoRCG achieves nearly a 13% improvement over EDM in unconditional generation for molecule stability, elevating it from one of the weakest models to the most powerful one in our comparison.
>
> To further enhance generation quality, one can certainly apply GeoRCG to a more advanced model, such as SemlaFlow,  and expecting even better performance, as shown in **Table 3 in the paper** and **Table 1 in the anonymous link**. This approach consistently improves result quality and pushes new SOTA results.
>
> We also respectfully invite you to refer to  **Tables 3 and 4 in the anonymous link** for our supplementary experiments in response to Reviewer eCAm, highlighting the benefits of GeoRCG in additional settings.
>
> ### **2. Is the incorporation of CFG and low-temperature sampling *fair*, and how does GeoRCG perform without them?**
> We very appreciate this suggestion and have investigated the impact of **totally disabling these components** under the most basic configuration (CFG = 0.0, Temperature = 1.0) to evaluate GeoRCG’s performance on both QM9 and DRUG datasets. The results are presented in **Table 2 in the anonymous link**. Our experiments demonstrate that GeoRCG **consistently enhances the base model** and remains competitive with more advanced methods.
>
> However, we would like to respectfully emphasize that CFG and low-temperature sampling (**note that low-temperature is for rep sampling**) are **fundamental to our approach**, as they represent how strongly we enforce representation guidance in the model and how varied the representation should be. Note that it is not that other molecular generative methods “did not use these techniques,” but rather that they “**cannot use these techniques**”: These methods lack representation conditioning and, as a result, are unable to control the conditioning scale or temperature of the representations. This means that the performance boost provided by these techniques **does not imply unfairness or “not brought by GeoRCG”.** In fact, in the computer vision domain [2, 3], these techniques are considered negligible and directly influence the effectiveness of the proposed methods.
>
> ### **3. Sampling time for the first-stage generation and GeoRCG (Semla)**
> We greatly appreciate your comment on this point. In **Table 1 of the anonymous link**, we provide measurements of the sampling steps and times for both stages.
> Notably, the **overhead** introduced by the first stage and by “adapting the SemlaFlow model to allow conditioning on this vector” **remains small**, even for a model as efficient as SemlaFlow: as evidenced in the “Rep. Sampling Time” column and “Molecule Sampling Time w/o CFG”. For more details, we respectfully refer you to our response to Reviewer eCAM under “Clarifications regarding sampling time and steps of SemlaFlow experiments.”
>
> ### **4. Additional pre-training dataset for UniMol [1]**
> In our setting, UniMol’s pre-training dataset includes not only GEOM-Drugs but also additional public or purchasable datasets used in its own pre-training settings, as described in its original paper [1].
>
> ---
> Once again, we sincerely appreciate your valuable comments and look forward to further discussion!
>
> [1] Zhou, Gengmo, et al. "Uni-mol: A universal 3d molecular representation learning framework."
>
> [2] Li, Tianhong, et al. "Return of unconditional generation: A self-supervised representation generation method."
>
> [3] Rombach, Robin, et al. "High-resolution image synthesis with latent diffusion models."

---

> > ### Comment · Reviewer_zxks · 2025-04-06
> >
> > Thank you for taking the time to answer my questions and add extensive additional experimental results. I will update my score appropriately.

---

> > > ### Author Response · Authors · 2025-04-07
> > >
> > > Dear Reviewer zxks,
> > >
> > > We sincerely thank you for taking the time to review our rebuttal and the additional materials. Your initial review and feedback are invaluable and greatly contributes to the improvement of our work. We will revise the paper accordingly to eliminate any potential ambiguities and include additional experimental evidence as you suggested.
> > >
> > > Thank you once again for your constructive input.
> > >
> > > Best regards,
> > >
> > > The Authors

---

### Official Review · Reviewer_5wZu · 2025-03-13

**Overall Recommendation:** 5

**Summary:**

This paper proposes a new framework, GeoRCG (Geometric-Representation-Conditioned Molecule Generation), to enhance the performance of molecular generation models. The proposed approach decomposes molecular generation into a two-step process as follows:

- **Geometric Representation Generation**: A pre-trained geometric encoder (such as Uni-Mol or Frad) is used to encode molecular geometric information into a compact representation.
- **Molecule Generation Based on Geometric Representation**: The representation obtained in the first step is used as a condition for a diffusion model (such as EDM or SemlaFlow) to generate molecules.

In diffusion model-based molecular generation, any feature vector can be used as a condition. The proposed method embeds molecules into a latent space using a molecular encoder that considers equivariance during training. A loss function is designed to enable sampling from this latent space for generation. As a result, the distribution in the latent space exhibits significantly better properties than the original molecular space, leading to improved molecule generation.

The main contributions of this work are as follows:

- **Improved molecular generation accuracy**: Achieves significantly better performance than existing methods (such as EDM and SemlaFlow) on the QM9 and GEOM-DRUG datasets.
- **Enhanced conditional molecular generation**: Demonstrates a 31% improvement over state-of-the-art methods in generating molecules while considering properties such as the HOMO-LUMO gap, polarity, and heat capacity.
- **Reduced computational cost**: Maintains nearly the same generation quality even when reducing the number of diffusion steps from 1,000 to 100.

**Claims And Evidence:**

- **Demonstrated Performance Improvement**: The proposed method outperforms state-of-the-art (SOTA) approaches such as GeoLDM, EDM-Bridge, GOAT, and GeoBFN. Notably, significant improvements in molecular stability and validity are observed.

- **Successful Conditional Generation**: Previous methods struggled with achieving target properties accurately. GeoRCG reduces property error by 31%, enabling more precise conditional molecular generation.

- **Reduced Computational Cost**: By incorporating geometric representations, the number of diffusion steps is reduced by 90% while maintaining the same generation quality as existing methods.

**Essential References Not Discussed:**

None

**Experimental Designs Or Analyses:**

I checked Section 4 for "Experiments", where the QM9 and GEOM-DRUG datasets are used for a detailed comparison of the generated molecules in terms of stability, validity, and property error. Although using only two datasets might be considered a limitation, **QM9** and **GEOM-DRUG** are representative benchmarks for evaluating **3D molecular graphs**. More importantly, the experiments comprehensively assess the proposed method from multiple perspectives, including **unconditional and conditional generation**.

**Methods And Evaluation Criteria:**

The proposed framework can broadly be considered a **latent generative model**, as it learns the data distribution in a latent space (first stage in this study) and reconstructs it into its original form using a decoder (second stage).

One key limitation of existing molecular generation models is that molecules inherently exist on a **low-dimensional manifold** (Mislow, 2012; De Bortoli, 2022; You et al., 2023), yet most approaches model them as distributions in a high-dimensional 3D space with \( N \times (3 + d) \) dimensions.

This work draws direct inspiration from **RCG (Li et al., 2023)**; however, RCG is designed for fixed-size, fixed-position image data and does not need to account for molecular-specific challenges such as Euclidean symmetry and permutation invariance. Similarly, **GraphRCG (Wang et al., 2024)** extends the RCG framework to **2D graph data**, whereas this study explicitly handles **3D geometric information with Euclidean symmetry**.

Furthermore, while **RCG (Li et al., 2023)** primarily focuses on empirical evaluations, this study generalizes the theoretical properties of **representation-conditioned diffusion models** for both **unconditional and conditional generation**, offering a more rigorous understanding of performance improvements.

**Other Comments Or Suggestions:**

None

**Other Strengths And Weaknesses:**

None

**Questions For Authors:**

None

**Relation To Broader Scientific Literature:**

While the approach of sampling from a latent space (as in RCG) is not new, to the best of my knowledge, it is novel in the field of **molecular generation**. Given its well-founded motivation, this method has the potential for **broad impact** in the field.

**Theoretical Claims:**

I briefly checked the content of Theorem 3.2 but did not rigorously check its proof in the supplementary information.

---

> ### Author Rebuttal · Authors · 2025-04-01
>
> Dear Reviewer 5wZu,
>
> We sincerely appreciate your expert and thoughtful review and are grateful for your recognition of our work.
>
> In response to the comments from the other reviewers, we have provided additional experimental evidence at https://anonymous.4open.science/r/rebuttal-8746 (Alternative link in case the previous one encounters issues: https://docs.google.com/document/d/e/2PACX-1vQFD87aQep2Q11albXeuuTKUr9HrrLIALteNrVVsbguQ92c_8ArXr_H43J8xv0WrnuRxbzwAOgBYsav/pub), which we believe will enhance the clarity and depth of our paper.
>
> We remain open to any further discussions or clarifications. Thank you once again for your valuable review!

---

> > ### Comment · Reviewer_5wZu · 2025-04-08
> >
> > Thank you for providing the additional experimental evidence. I agree that this information will enhance the clarity and depth of our paper. Nice and interesting work!

---

### Official Review · Reviewer_eCAm · 2025-03-19

**Overall Recommendation:** 4

**Summary:**

The paper presents GeoRCG, a novel framework for 3D small molecule generation, applicable to both unconditional and conditional settings. The key innovation is a two-step generation process: first, generating a geometric representation, then sampling the 3D molecular structure conditioned on this representation. This approach aims to simplify the generative process, improve sample quality, and speed up molecule generation.

Key benefits in downstream evaluation include:

- Reduced computational cost in comparison to diffusion-based methods.
- Comparative evaluation against state-of-the-art methods, showing significant improvements in conditional generation tasks.
- Theoretical bounds for the proposed diffusion setup.

## Update after rebuttal

I have reviewed the additional tables and experiments provided by the authors, and they have satisfactorily address my concerns about fair comparisons and missing metrics. I suggest that authors revise their claims to provide more context in the final version, and incorporate the additional tables in the main paper/appendix. I have raised my rating to 4 (Accept).

**Claims And Evidence:**

The proposed claims needs some revision to provide more context:
- Claim: The proposed method significantly improves molecule generation quality.
    - These results hold only for the setup of conditional generation, the results are comparable to other SOTA models in unconditional setup. This should be clarified.

- Claim: The proposed method achieves faster generation with fewer steps.
   - Speed-up should only be claimed in comparison to other diffusion models, as some other methods (e.g., SemlaFlow) using flow matching use the same number of steps, and could arguably sample faster due to a single step sampling procedure instead of two.

- Claim: The paper reports an average 31% improvement over SOTA baselines.
     - The comparison with SemlaFlow is missing, and the results might not be as strong if SemlaFlow is also trained in conditional setup and using the same number of sampling steps. However, this is justifiable since SemlaFlow paper does not report conditional generation results.

**Essential References Not Discussed:**

The paper discusses all the relevant essential references.

**Experimental Designs Or Analyses:**

The experimental design is overall justified, however there are missing details in the evaluation and claims as described in the previous section.

**Methods And Evaluation Criteria:**

- The overall proposed framework is reasonable for the problem. Representation generation prior to sampling is unexplored in molecular generation context, and the design of the paper is well motivated.
- The main evaluation datasets are QM9 and Geom-Drugs as is the standard practice for papers in this area.
- The evaluation primarily uses validity, stability, and uniqueness metrics.:
     - Energy and strain metrics are missing from Table 2.
     - It is recommended that energy and strain **per atom** metrics are used as is the standard practice.
 - Clarity in evaluation setup and comparisons:
    - Table 3 and Table 4 should be integrated into Table 2 and all the methods should be compared together to provide accurate context for the improvements. The claimed speedups only hold when flow based methods such as SemlaFlow are also added to the table.
- Comparison with SemlaFlow needs more depth, particularly regarding:
    - Can SemlaFlow also be used for conditional generation ?
    - It should be clarified if both the proposed method and SemlaFlow have similar computational costs.
    -  Whether representation learning provides a real benefit over SemlaFlow

**Other Comments Or Suggestions:**

The clarity of the paper can be improved. Several key modeling details are available in the Appendix, the readers would benefit from providing more information in the main paper. The evaluation and results are not well presented and several tables can be integrated to provide a clearer picture of head-on-head comparisons.
The figure needs improvement and does not provide a good picture in describing the method.

**Other Strengths And Weaknesses:**

- Strengths:

    - Novel approach: The two-step generation process is well motivated.
    - Strong empirical results: Significant improvements in conditional generation.
    - Efficient generation: The model reduces diffusion steps without quality loss.

- Weaknesses:
    - Missing clarity in evaluation: Too many tables, and all the methods are not compared head-on.
    - Major claims require more context, as stated in the previous section.

**Questions For Authors:**

- Can SemlaFlow be used in conditional settings?
- Could the authors comment on the sampling steps for SemlaFlow and the overall generation time in comparison ?
- I would like to review all the methods presented in Tables 1,3, and 4 compared against each other, by clearly indicating the number of samplings steps for each and other important metrics (including stability and energy).

I would consider raising my rating if the above concerns are satisfactorily addressed.

**Relation To Broader Scientific Literature:**

The paper addresses the problem of conditional and unconditional 3D small molecule generation. Other works in this area primarily develop diffusion/flow based generative modeling for learning the molecular distributions and subsequently sampling. Contrary to previous works, this paper proposes a 2-step procedure, for first generating a representation and subsequently sampling the molecule.

**Theoretical Claims:**

I did not carefully check the proofs, but the theoretical claims are mostly derivative from the previous works.

---

> ### Author Rebuttal · Authors · 2025-04-01
>
> Dear Reviewer eCAm,
>
> We sincerely appreciate your thorough and insightful review! Below, we address each of your comments in detail. **We have provided additional tables in an anonymous link: https://anonymous.4open.science/r/rebuttal-8746. (Please refer to our response to Reviewer zxks for an *alternative link* in case the previous one encounters issues.)**
>
> ---
>
> ###  **1. Claim Clarifications and Conditional Experiments for SemlaFlow**
> We acknowledge that some claims need additional context for clarity, and are committed to modifying them in the next revision. Here are brief clarifications:
> - In Claim 1, 2, we refer to “improvements” and "faster" **over base models**, where our GeoRCG consistently enhances the performance of base models.
> - Regarding Claim 3, we have conducted our own experiments for SemlaFlow's conditional generation for a more comprehensive comparison. Please refer to **Table 4 in the link.** The results demonstrate SemlaFlow’s superior performance in this setting; however, **GeoRCG consistently enhances SemlaFlow’s performance in conditional generation.**
> ### **2. Comparing all the methods head-on.**
> We clarify that we have split the experimental results for several important reason:
> - **Differences in dataset processing and evaluation:** The models in Table 3 were trained with different processing pipelines than those in Table 1 (e.g., Table 3 used the 5 lowest-energy conformations per structure for DRUG, while Table 1 used the 30 lowest). Additionally, dataset splits and evaluation criteria (e.g., allowable valences, molecule sanitization) differ significantly.
> - **Differences in model architecture, i.e., 2D&3D vs. 3D-only:** Models in Table 3, such as SemlaFlow, jointly learn both 3D conformations and 2D bonds, while 3D-only models like EDM focus solely on 3D conformations. **Direct comparison is unfair** since metrics like validity and stability depend on 2D graphs, making 2D&3D models generally perform better.
> - **Difference in evaluation focus:** Tables 4 and 5 report performance with fewer sampling steps, focusing on efficiency, while Table 1 shows best-case performance. However, **for a more comprehensive comparison of 3D-only models, we have reorganized the results into Table 5 in the link** respectfully for your review.
> ### **3. Why we did not include energy and strain metrics in Table 1, 2, 4, 5?**
> We acknowledge the absence of these metrics, and would like to provide the following clarifications:
> - Energy and strain metrics are introduced by SemlaFlow due to the **lack of effective 3D metrics for evaluating the 3D conformations generated by these 2D&3D models**, reasons as stated above. However, for the 3D-only methods presented in Tables 1, 2, 4, 5, the stability, validity, and property MAE metrics **already provide a reflection of the quality of the generated 3D conformations**, since the 2D bonds or properties used in these calculations are inferred from the generated 3D conformations.
> - For completeness, we evaluate our method along with selected baseline models (EDM, GeoLDM) on energy/strain metrics, please see **Table 3 in the link**. Notably, **GeoRCG continues to show consistent improvements under these metrics as well**.
> - We appreciate your suggestion regarding “Per Atom Metrics” and have added per atom metrics. Please see **Table 1, 3 in the link.**
> ### **4. Clarifications regarding sampling time and steps of SemlaFlow experiments.**
> Thank you for your valuable comment regarding this! In response, we have annotated the sampling steps and times in **Table 1 in the link**. We would like to highlight several key points:
> - GeoRCG consistently improves upon SemlaFlow **under the same sampling steps.**
> - The first stage (rep sampling) incurs significantly less computational cost than the second stage, even for SemlaFlow, which is already highly efficient in molecule generation.
> - GeoRCG (Semla) takes about twice the time of SemlaFlow with the same sampling steps, mainly due to classifier-free guidance, which doubles the batch size to enable guidance [1]. However, we argue that:
>     - Even with **half the sampling steps** (roughly the same time), GeoRCG **still outperforms base models** (Tables 4 and 5 in the main paper, and Table 1 in the link).
>     - Even **without CFG** (same steps → similar time), GeoRCG often surpasses its base model (Table 2 of the link).
>     - With advancements in GPUs, the overhead from CFG will become less significant due to parallel acceleration, while the sequential nature of generative models will continue to be a limiting factor.
>
> In summary, GeoRCG holds its efficiency advantage for SemlaFlow, too. We will add a detailed discussion of SemlaFlow’s sampling time and include a more thorough analysis of the impact of CFG in the revision.
>
> ---
>
> Once again, we sincerely appreciate your valuable comments and look forward to further discussion!
>
> [1] Ho, Jonathan, et al. "Classifier-free diffusion guidance."

---

> > ### Comment · Reviewer_eCAm · 2025-04-01
> >
> > Thanks for the detailed rebuttal. I have reviewed the additional tables provided by the authors, and they have satisfactorily address my concerns about fair comparisons and missing metrics. I suggest that authors revise their claims to provide more context in the final version, and incorporate the additional tables in the main paper/appendix. I have raised my rating.

---

> > > ### Author Response · Authors · 2025-04-02
> > >
> > > Dear Reviewer eCAm,
> > >
> > > Thank you for taking the time to review our rebuttal and the additional materials. We truly appreciate your thoughtful and timely feedback, and we’re glad to hear that your concerns have been addressed.
> > >
> > > We’re also grateful for your review, which has helped us improve the depth and clarity of the paper. As suggested, we will revise our claims in the final version to provide better context and will incorporate the additional tables into the main paper or appendix as appropriate.
> > >
> > > Thank you again for your constructive input.
> > >
> > > Best regards,
> > >
> > > The Authors

---

### Decision · Program_Chairs · 2025-05-01

**Decision:**

Accept (spotlight poster)

**Comment:**

This submission presents an approach to decompose molecular generation into two stages. In the first stage a geometric representation is generated and in the second a molecule is generated conditioned on the geometric representation. The paper is well written and the experimental evaluation supports the claims of the paper. Reviewers collectively agreed on the value of this submission to the ICML community.